# Non-linear developmental trajectory of electrical phenotype in rat substantia nigra pars compacta dopaminergic neurons

**Martial A Dufour[1,2], Adele Woodhouse[1,2†], Julien Amendola[1,2], Jean-Marc Goaillard[1,2]\***

[1]Inserm UMR 1072, Faculté de Médecine Secteur Nord, Université de la Méditerranée, Marseille, France; [2]Aix-Marseille Université, Marseille, France

**Abstract** Neurons have complex electrophysiological properties, however, it is often difficult to determine which properties are the most relevant to neuronal function. By combining current-clamp measurements of electrophysiological properties with multi-variate analysis (hierarchical clustering, principal component analysis), we were able to characterize the postnatal development of substantia nigra dopaminergic neurons' electrical phenotype in an unbiased manner, such that subtle changes in phenotype could be analyzed. We show that the intrinsic electrical phenotype of these neurons follows a non-linear trajectory reaching maturity by postnatal day 14, with two developmental transitions occurring between postnatal days 3–5 and 9–11. This approach also predicted which parameters play a critical role in phenotypic variation, enabling us to determine (using pharmacology, dynamic-clamp) that changes in the leak, sodium and calcium-activated potassium currents are central to these two developmental transitions. This analysis enables an unbiased definition of neuronal type/phenotype that is applicable to a range of research questions.

**\*For correspondence:** jean-marc.goaillard@univ-amu.fr

**Present address:** †Wicking Dementia and Research Education Centre, University of Tasmania, Hobart, Australia

**Competing interests:** The authors declare that no competing interests exist.

**Reviewing editor**: Ronald L Calabrese, Emory University, United States

## Introduction

The morphology and assortment of voltage-dependent and voltage-independent conductances displayed by one particular neuronal type provide it with specific passive and active properties (*Johnston and Wu, 1995*; *Hille, 2001*). In turn, passive and active properties define the way the neuron produces or processes information, that is, its electrical phenotype. An important question we are facing as neurophysiologists is how to characterize the electrical phenotype of a neuronal type in the most unbiased manner, such that we can then study subtle variations in this phenotype under varying conditions (development, perturbations, disease). In the current study, we propose an approach based on the combination of current-clamp measurements and multi-dimensional analysis of the measured electrophysiological properties, which allowed us to characterize and precisely quantify developmental changes in the electrical phenotype of dopaminergic neurons of the substantia nigra pars compacta.

In vivo, mature SNc dopaminergic neurons can display regular tonic, irregular or bursting activity depending on the behavioral context (*Grace and Bunney, 1984a*, *1984b*; *Tepper et al., 1990*; *Kitai et al., 1999*; *Grace et al., 2007*). These variations in activity patterns are associated with a modulation of dopamine release such that, for instance, immediate early gene activation is triggered only by bursting patterns of activity (*Kitai et al., 1999*). From a biophysical point of view, regular tonic activity seems to rely mainly on SNc dopaminergic neuron intrinsic conductances while both irregular and bursting patterns necessitate the release of various neurotransmitters (glutamate, acetylcholine) by SNc synaptic inputs (*Kitai et al., 1999*; *Grace et al., 2007*). In vitro however, mature SNc

**eLife digest** The brain contains hundreds of types of neurons, which differ in size and shape, and also in the chemicals that they use to communicate with each other. However, one thing all neurons have in common is that they all carry electrical signals that depend on the flow of ions through specialized channels in the membranes that surround each neuron. Nevertheless, the number and identity of these channels also vary markedly from one type of neuron to the next.

This biophysical diversity underlies a variety of complex patterns of electrical activity observed in different types of neurons. This complexity often makes it difficult to pinpoint which of the myriad features of a neuron are most important for determining its function, and which ones are most affected by processes such as aging or disease. To address this problem, Dufour et al. have devised an approach for characterizing the electrical properties of neurons in a systematic manner, in order to generate a 'phenotypic' profile for individual types of neurons.

Neurons from a region of the brain called the substantia nigra pars compacta were chosen for the study. These neurons are known for the fact that they are among the first to degenerate in Parkinson's disease. Using electrodes to record from slices of rat brain, a total of 16 electrical properties were measured for each neuron, including how often each cell 'fired' and how variable its firing pattern was. The experiments were repeated using brain slices from rats aged between 2 and 29 days, and the data from more than 300 neurons were then analyzed using statistical techniques designed to identify groups of features that change together over time.

The analysis revealed that the cells could be grouped into three developmental stages, separated by two transitions: one occurring around days 3–5 and another around days 9–11. This was confirmed by experiments showing that cells could be made to revert back to an earlier stage by applying chemicals and electrical currents to reverse the changes that had occurred during development.

While the current study has provided insights into the postnatal development of one particular class of neurons in the substantia nigra, the approach could in principle be applied to any type of neuron. The detailed profile that is obtained will make it easier to identify subtle changes in neurons in response to development, aging, or disease.

dopaminergic neurons mainly display a regular tonic (also called pacemaker) firing behavior (*Grace and Onn, 1989*; *Liss et al., 2001*; *Seutin et al., 2001*; *Puopolo et al., 2007*; *Guzman et al., 2009*; *Putzier et al., 2009b*; *Tateno and Robinson, 2011*; *Amendola et al., 2012*). The irregular pattern of spontaneous activity is observed in immature SNc DA neurons in vitro (postnatal days 9–16, P9–P16) and seems to be due to transient spontaneous activation of T-type calcium channels (*Seutin et al., 2000*; *Cui et al., 2004*) and subsequent activation of calcium-activated potassium channels (SK) (*Seutin et al., 1998*; *Wolfart and Roeper, 2002*; *Cui et al., 2004*). Finally, the spontaneous bursting pattern of activity is usually absent from mature SNc DA neurons in vitro, but can be observed in juvenile neurons (P15–P21) and depends on the activation of NMDA receptors (*Mereu et al., 1997*), or can be promoted by pharmacological blockade of T-type calcium channels and SK potassium channels (*Wolfart and Roeper, 2002*).

Although these different studies suggest that the activity pattern of SNc dopaminergic neurons changes during the first three postnatal weeks, involving modifications in both intrinsic and synaptic input properties, our knowledge of the precise timecourse of their electrophysiological development is still fragmented. In the current study, we have performed a detailed analysis of the development of the intrinsic properties of SNc dopaminergic neurons over the first four postnatal weeks (from P2 to P29). We measured 16 electrophysiological parameters, exhaustively characterizing the passive and active properties of SNc dopaminergic neurons. Applying multi-variate statistical analyses (agglomerative hierarchical clustering and principal component analysis) to the measured electrophysiological parameters revealed that the acquisition of mature regular pacemaking involves biphasic changes occurring mainly during the first two postnatal weeks: intrinsic electrophysiological maturity is essentially reached by the end of the second postnatal week. In addition, this type of analysis allowed us to determine the participation of specific ion currents (leak, sodium and calcium-activated potassium currents) to changes in the global electrical phenotype. This study provides the first comprehensive analysis of postnatal development of the intrinsic properties of SNc dopaminergic neurons and

demonstrates the utility of high-dimensional electrophysiological characterization associated with multi-variate analysis methods to precisely define quantitative changes in electrical phenotype and to investigate the underlying biophysical mechanisms.

## Results

In order to define the postnatal development of the intrinsic electrophysiological properties of SNc dopaminergic neurons, we recorded from SNc dopaminergic neurons in acute midbrain slices made from rats aged between 2 and 29 postnatal days (P2 to P29) in the presence of blockers of glutamatergic and GABAergic synaptic activities (Kynurenate and Picrotoxin). SNc dopaminergic neurons were identified based on their location, size, and morphology (*Figure 1A*) and their characteristic electrophysiological properties, including a slow AP and the presence of a large sag in response to hyperpolarizing current pulses (*Figure 1B*). Independent of the developmental stage, these electrophysiological features allowed reliable identification of SNc dopaminergic neurons vs SNc GABAergic neurons: when *post hoc* tyrosine hydroxylase immunolabeling was performed, dopaminergic identity was confirmed in 100% of the cases (n = 121, see *Figure 1A*).

### Developmental evolution of spontaneous activity patterns

We first analyzed the spontaneous activity patterns displayed by SNc dopaminergic neurons, using two simple measures capturing the general features of activity: the averaged InterSpike Interval ($ISI_{avg}$), and the coefficient of variation of the ISI ($CV_{ISI}$), which is proportional to the irregularity of firing (*Figure 2A*). While $ISI_{avg}$ was found to be stable from P2 to P29 (*Figure 2B,C*), $CV_{ISI}$ strongly decreased over the first two postnatal weeks, reaching a stable value by P10–11 (*Figure 2B,D*, *Table 1*, *Figure 3*). The decrease in $CV_{ISI}$ was correlated with changes in firing pattern, with high $CV_{ISI}$ values associated with a bursting pattern, intermediate values associated with irregular tonic firing and low values with regular tonic firing (*Figure 2B*). In fact, $CV_{ISI}$ threshold values of 20% and 80% were found to reliably separate these three types of firing patterns (*Figure 2D,E*). Using these thresholds, the proportions of high $CV_{ISI}$ (bursting cells), medium $CV_{ISI}$ (irregular cells), and low $CV_{ISI}$ (regular cells) were calculated for each developmental stage (*Figure 2E*): all neurons were found to be bursters at P2–3, most neurons were irregular between P5 and P11, while pacemaker neurons became predominant after P12 and were the only type of neurons encountered after P21 (*Figure 2E*).

Several studies have suggested that bursting and irregular firing patterns are controlled in part by the calcium-activated potassium channels (SK1-3) responsible for the medium afterhyperpolarization (AHP) following APs (*Seutin et al., 1998*; *Wolfart et al., 2001*; *Wolfart and Roeper, 2002*; *Cui et al., 2004*; *Johnson and Wu, 2004*; *Vandecasteele et al., 2011*). In SNc dopaminergic neurons, it was demonstrated that SK channels determine the regularity of spontaneous activity (*Seutin et al., 1998*; *Wolfart et al., 2001*; *Cui et al., 2004*; *Vandecasteele et al., 2011*), and that AHP developmental increase correlates with the disappearance of irregular firing (*Vandecasteele et al., 2011*). Since our recordings were performed using 10 mM EGTA in the intracellular solution, we wondered whether the resulting calcium chelation could inhibit SK channel activation and thus modify spontaneous activity patterns (favoring irregular or bursting firing patterns). We therefore performed two types of control experiments. In the first series of experiments, we reduced the intracellular EGTA concentration to 0.5 mM (*Figure 2—figure supplement 1A–C*). In spite of the reduced calcium chelation, all firing patterns observed in 10 mM EGTA were also present in 0.5 mM EGTA (*Figure 2—figure supplement 1A*). Moreover, the average $CV_{ISI}$ values were not statistically different between both EGTA concentrations (p = 0.201, n = 80 for 10 mM, n = 34 for 0.5 mM, unpaired *t* test; *Figure 2—figure supplement 1B*) when measured in neurons of the same developmental stage (P18–P22). Since the AHP can be influenced by calcium chelation and is involved in defining $CV_{ISI}$, we also verified that AHP amplitude was not significantly modified by changes in EGTA concentration (*Figure 2—figure supplement 1C*): the mean AHP amplitude was not statistically different between both EGTA concentrations in P18–22 neurons (p = 0.129, n = 100 for 10 mM, n = 28 for 0.5 mM, unpaired *t* test; *Figure 2—figure supplement 1C*). As 0.5 mM EGTA could still be considered as a calcium chelation condition that could significantly modify firing (*Grace and Bunney, 1984b*), we also verified that the three types of firing observed in whole-cell recordings in 0.5 mM EGTA or 10 mM EGTA were also present in cell-attached recordings. As demonstrated in *Figure 2—figure supplement 1D*, the firing patterns observed in 0.5 mM EGTA whole-cell recordings were found to be qualitatively similar to the firing patterns observed in cell-attached recordings in the same neurons before going whole-cell: regular, irregular,

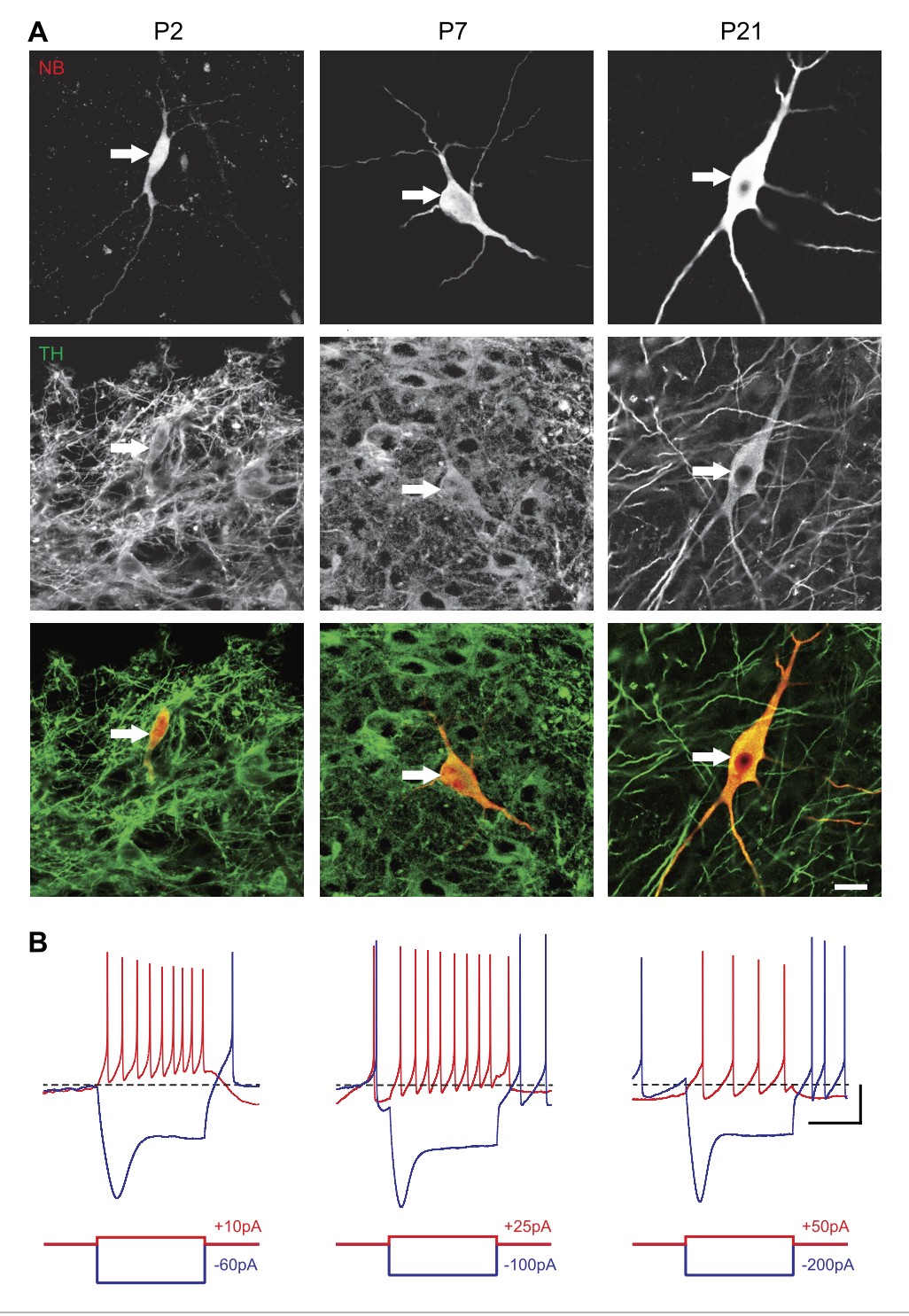

**Figure 1**. Identification of substantia nigra pars compacta dopaminergic neurons during postnatal development. (**A**), top, fluorescent streptavidin labeling of P2 (left), P7 (center) and P21 (right) neurons filled with neurobiotin (NB). Middle, tyrosine hydroxylase (TH) immunolabeling of the same neurons. Bottom, merged images showing the NB (red) and TH (green) labeling, confirming the dopaminergic nature of the recorded neurons. (**B**), characteristic voltage traces obtained from P2 (top left), P7 (top center) and P21 (top right) subtantia nigra pars compacta dopaminergic neurons in response to hyperpolarizing (blue) and depolarizing (red) current pulses (bottom traces). At each developmental stage, dopaminergic neurons displayed consistent TH labeling (**A**), as well as a typical sag

*Figure 1. Continued on next page*

*Figure 1. Continued*

in response to hyperpolarizing current pulses (**B**) and a broad AP (not shown). Scale bars: **A**, 10 μm; **B**, vertical 20 mV, horizontal 500 ms. The horizontal dotted line in **B** indicates −60 mV.

and bursting patterns were observed in cell-attached recordings. To ensure that whole-cell recorded firing patterns were also quantitatively similar to cell-attached recorded firing patterns, we compared the frequency of regular tonic firing in a subset of 33 neurons (P16–P23) that were recorded both in cell-attached (before breaking into the neuron) and in whole-cell 10 mM EGTA (*Figure 2—figure supplement 1E*). This analysis revealed that $ISI_{avg}$ was not altered by the whole-cell recording configuration (p = 0.898, n = 33, paired *t* test; *Figure 2—figure supplement 1E*). Therefore, the activity pattern and activity level are not significantly altered by the recording conditions, and the developmental transition between the different types of firing patterns cannot be explained by the recording conditions.

## Developmental evolution of passive properties

We then analyzed the passive membrane properties of SNc dopaminergic neurons, including the input resistance ($R_{in}$), the membrane capacitance ($C_m$), and the membrane time constant ($\tau_m = R_{in} \times C_m$). $R_{in}$, $C_m$, and $\tau_m$ values were extracted from mono-exponential fits of the voltage responses to small hyperpolarizing current pulses applied around resting membrane potential (*Figure 4A,B*). Except for P2–3 where $R_{in}$ values were particularly high (*Figure 4C*, *Table 1*), $R_{in}$ and $C_m$ displayed almost symmetrical changes over the first four postnatal weeks: $R_{in}$ decreased while $C_m$ increased (*Figure 3*, *Figure 4C,D*, *Table 1*), such that $\tau_m$ remained fairly constant over the same timeframe (*Figure 3*, *Figure 4E*, *Table 1*). These data suggest that, after P2–3, membrane surface area increases but that leak conductances' density stays fairly constant, such that the measured input conductance ($1/R_{in}$) scales with $C_m$ (since $C_m$ scales with membrane surface area).

## Developmental evolution of post-inhibitory rebound properties

SNc dopaminergic neurons are characterized by a specific response to hyperpolarizing current pulses consisting of a large sag during the hyperpolarization due to the activation of $I_H$, and a biphasic post-inhibitory rebound involving both $I_H$ and the transient potassium current $I_A$ (*Amendola et al., 2012*). We analyzed the response to hyperpolarization by measuring both sag amplitude and rebound delay (*Figure 5A*). In spite of the increase in amplitude of $I_H$ that has been described during postnatal development in SNc dopaminergic neurons (*Washio et al., 1999*), we found that both sag amplitude and rebound delay remained constant during the first four postnatal weeks (*Figure 3*, *Figure 5B,C*, *Table 1*), suggesting that the density of $I_A$ and $I_H$ remained fairly stable during this developmental timeframe.

## Developmental evolution of action potential properties

SNc dopaminergic neurons are also characterized by a slow action potential (AP) (*Grace, 1990*) that relies on the activation of multiple currents including calcium, transient sodium, and A-type, delayed rectifier and calcium-activated potassium currents (*Bean, 2007*; *Puopolo et al., 2007*). In order to describe the developmental evolution of AP shape, we measured six properties defining the different phases of the AP recorded during spontaneous activity (*Figure 6A*): AP threshold, AP rise slope, AP decay slope, AP half-width, AP amplitude, and AHP amplitude. Except for AP threshold, which was found to not vary significantly during the first four postnatal weeks (*Figure 6C*, *Table 1*, *Figure 3*), most AP features were found to present significant modifications over the same developmental timeframe (*Figure 3*, *Figure 6B–H*, *Table 1*). Some of these properties presented monophasic changes, such as AP decay slope, which decreased over time (*Figure 6B,F*, *Table 1*), or AHP amplitude, which increased over time (*Figure 6B,H*, *Table 1*). Surprisingly though, AP amplitude, rise slope and half-width displayed biphasic changes during the same developmental timeframe, with the first change between P2–3 and P5 and a second opposite change between P8–9 and P13 (*Figure 6B,D,E,G*, *Table 1*): AP amplitude and AP rise slope first decreased and then increased, while AP half-width presented an increase in value followed by a decrease. These changes are clearly visible on the recordings of APs from P3, P6, and P21 animals (*Figure 6B*). Therefore, the different AP properties measured exhibit distinct evolutions, which suggest heterogeneous changes in expression or gating properties of the different types of ion channels involved in AP shape definition.

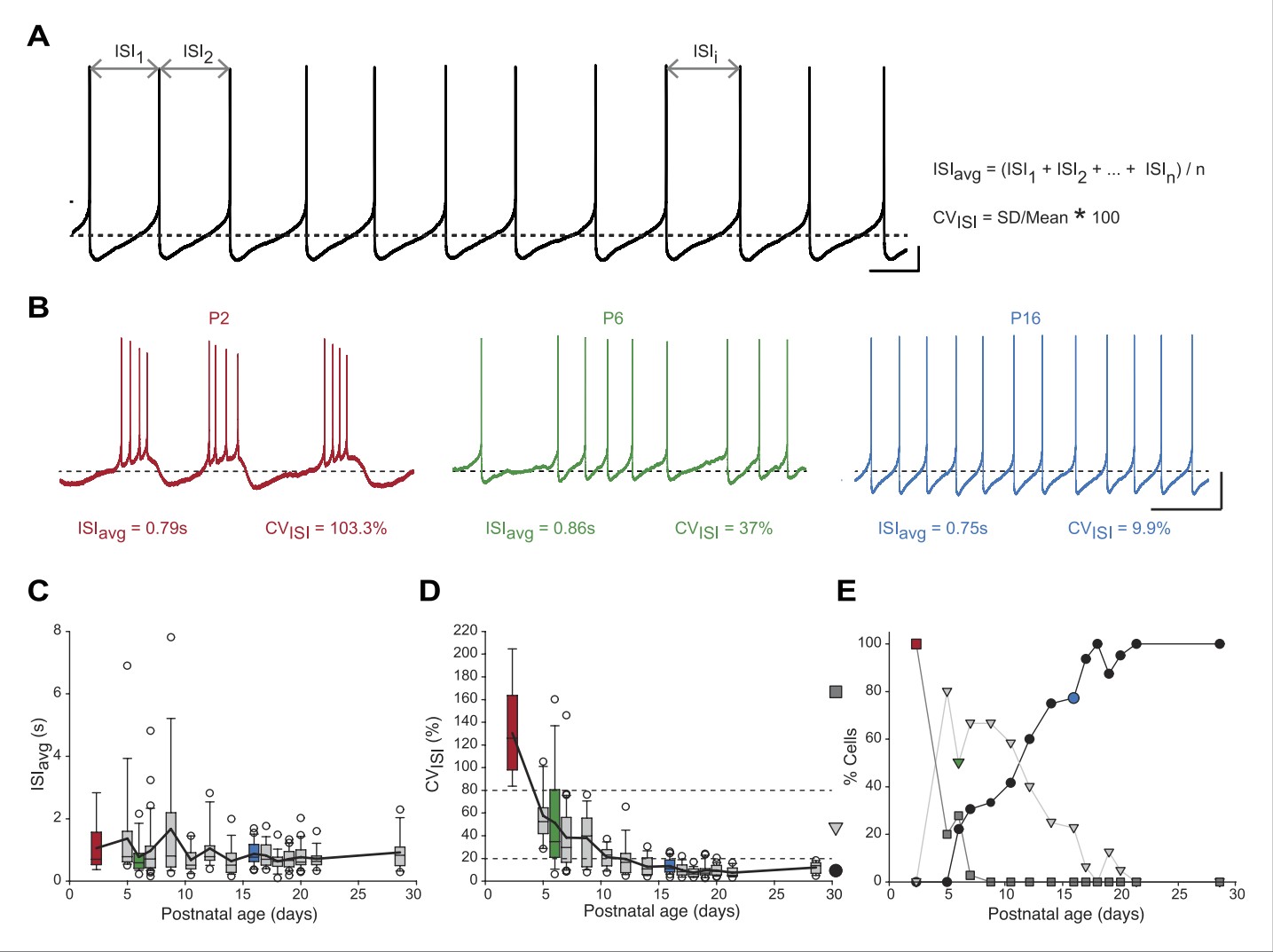

**Figure 2**. Postnatal evolution of spontaneous activity patterns in substantia nigra pars compacta dopaminergic neurons. (**A**), typical voltage recording from a regular pacemaker dopaminergic neuron depicting the parameters extracted to characterize spontaneous activity patterns. Interspike intervals ($ISI_1$, $ISI_2$…$ISI_i$…$ISI_n$) were averaged to calculate the $ISI_{avg}$ while the coefficient of variation of the n ISIs ($CV_{ISI}$) was computed from the standard deviation (SD) of the n ISIs and $ISI_{avg}$. (**B**), voltage traces showing the bursting (high $CV_{ISI}$, left, red), irregular (moderate $CV_{ISI}$, center, green) and regular (low $CV_{ISI}$, right, blue) patterns of spontaneous activity observed during the first four postnatal weeks. (**C**), box and whisker plot representing $ISI_{avg}$ vs postnatal age. (**D**), box and whisker plot representing $CV_{ISI}$ vs postnatal age. Two $CV_{ISI}$ threshold values (20 and 80%, horizontal dotted lines) separated three classes of activity patterns: low CV ($CV_{ISI} < 20\%$, black circle), medium CV ($20\% < CV_{ISI} < 80\%$, light gray triangle), high CV ($CV_{ISI} > 80\%$, dark gray square). The dark gray square, light gray triangle and black circle on the right indicate the symbols used to represent the three different CV classes (high CV, medium CV and low CV, respectively) in panel **E**. (**E**), line and scatter plot representing the evolution of the percentages of high CV (dark gray square), medium CV (light gray triangle) and low CV (black circle) activity patterns vs postnatal age. Scale bars: **A**, vertical 10 mV, horizontal 500 ms; **B**, vertical 20 mV, horizontal 2 s. Horizontal dotted lines in **A** and **B** indicate −60 mV. For all box and whisker plots, boxes represent the median, first and third quartile, error bars correspond to 10 and 90%, the thick line corresponds to the mean, and all outliers are represented. Colored boxes and symbols in **C**, **D** and **E** match the age and type of firing of the colored traces presented in **B** (red for P2, bursting; green for P6, irregular; blue for regular, P16).

The following figure supplement is available for figure 2:

**Figure supplement 1**. Lack of effect of the recording condition on the activity pattern of substantia nigra pars compacta dopaminergic neurons.

## Developmental evolution of membrane excitability

In order to determine changes in membrane excitability, we measured the frequency/current response of SNc dopaminergic neurons in response to incremental depolarizing current pulses (*Figure 7A*). Since SNc dopaminergic neurons have been shown to display spike frequency adaptation (SFA) during

**Table 1.** Descriptive statistics of the 16 electrophysiological parameters measured on substantia nigra pars compacta dopaminergic neurons across postnatal development

| Age | ISI$_{avg}$ (s) Mean | SD | N | CV$_{ISI}$ (%) | | | τ$_m$ (ms) | | | R$_{in}$ (MOhm) | | | C$_m$ (pF) | | | Sag (mV) | | | Rebound (ms) | | | AP threshold (mV) | | |
|---|---|---|---|---|---|---|---|---|---|---|---|---|---|---|---|---|---|---|---|---|---|---|---|---|
| P2–P3 | 1.06 | 0.84 | 9 | 130.5 | 40.1 | 9 | 200.3 | 8.0 | 5 | 1642 | 420 | 5 | 129 | 34 | 5 | 28.2 | 4.0 | 9 | 465 | 134 | 6 | −43.9 | 3.3 | 9 |
| P5 | 1.36 | 1.60 | 15 | 57.6 | 22.6 | 15 | 86.9 | 13.7 | 11 | 642 | 163 | 11 | 146 | 52 | 11 | 29.4 | 5.7 | 14 | 339 | 188 | 12 | −40.3 | 4.5 | 15 |
| P6 | 0.78 | 0.56 | 18 | 51.5 | 43.0 | 18 | 70.3 | 23.9 | 18 | 612 | 170 | 18 | 118 | 32 | 18 | 30.2 | 7.2 | 18 | 316 | 127 | 18 | −43.1 | 4.3 | 19 |
| P7 | 0.97 | 0.93 | 36 | 38.4 | 28.0 | 36 | 89.3 | 24.4 | 25 | 650 | 174 | 26 | 143 | 38 | 25 | 31.2 | 5.2 | 25 | 249 | 89 | 19 | −39.8 | 4.0 | 35 |
| P8–P9 | 1.67 | 1.95 | 18 | 38.0 | 23.2 | 18 | 90.9 | 38.4 | 21 | 576 | 268 | 21 | 200 | 141 | 21 | 29.6 | 6.2 | 17 | 401 | 212 | 15 | −42.8 | 4.3 | 20 |
| P10–P11 | 0.68 | 0.41 | 12 | 21.1 | 8.6 | 12 | 80.6 | 28.6 | 10 | 327 | 210 | 10 | 281 | 109 | 10 | 33.3 | 4.2 | 10 | 330 | 165 | 10 | −41.7 | 4.2 | 11 |
| P12–P13 | 1.04 | 0.67 | 15 | 19.6 | 15.1 | 15 | 87.9 | 32.7 | 17 | 369 | 166 | 17 | 297 | 226 | 17 | 32.4 | 3.7 | 16 | 378 | 341 | 16 | −45.4 | 3.7 | 18 |
| P14 | 0.64 | 0.48 | 16 | 12.8 | 8.3 | 16 | 92.3 | 44.0 | 14 | 460 | 192 | 14 | 205 | 68 | 14 | 35.6 | 2.9 | 10 | 489 | 448 | 9 | −44.2 | 2.3 | 16 |
| P15–P16 | 0.88 | 0.38 | 22 | 13.3 | 6.6 | 22 | 77.8 | 28.5 | 19 | 325 | 117 | 19 | 245 | 67 | 19 | 27.9 | 6.5 | 33 | 340 | 207 | 30 | −44.4 | 3.4 | 30 |
| P17 | 0.82 | 0.40 | 16 | 9.9 | 5.6 | 16 | 83.5 | 25.7 | 20 | 323 | 109 | 22 | 283 | 160 | 20 | 28.1 | 7.4 | 35 | 375 | 270 | 26 | −44.6 | 3.5 | 28 |
| P18 | 0.63 | 0.30 | 18 | 7.6 | 4.0 | 18 | 89.0 | 24.2 | 18 | 340 | 118 | 18 | 280 | 86 | 18 | 29.5 | 4.2 | 22 | 433 | 222 | 20 | −41.6 | 3.0 | 21 |
| P19 | 0.71 | 0.32 | 24 | 9.6 | 6.1 | 24 | 89.9 | 28.1 | 30 | 352 | 103 | 30 | 270 | 106 | 30 | 28.6 | 6.2 | 38 | 456 | 278 | 32 | −44.2 | 4.1 | 33 |
| P20 | 0.77 | 0.42 | 21 | 9.3 | 4.8 | 21 | 88.5 | 41.8 | 22 | 375 | 116 | 23 | 231 | 66 | 22 | 28.5 | 8.3 | 24 | 572 | 493 | 22 | −44.3 | 4.4 | 24 |
| P21–P23 | 0.72 | 0.32 | 19 | 7.5 | 4.5 | 19 | 93.1 | 48.2 | 21 | 319 | 115 | 23 | 293 | 128 | 21 | 27.5 | 6.6 | 23 | 388 | 254 | 22 | −43.9 | 4.0 | 24 |
| P28–P29 | 0.92 | 0.54 | 12 | 12.1 | 4.7 | 12 | 148.3 | 42.6 | 10 | 472 | 88 | 10 | 316 | 83 | 10 | 34.1 | 4.5 | 13 | 736 | 375 | 10 | −42.8 | 3.0 | 12 |

| Age | AP amplitude (mV) | | | AP half-width (ms) | | | AP rise slope (mV/ms) | | | AP decay slope (mV/ms) | | | AHP (mV) | | | Gain start (Hz/100pA) | | | Gain end (Hz/100pA) | | | SFA index | | |
|---|---|---|---|---|---|---|---|---|---|---|---|---|---|---|---|---|---|---|---|---|---|---|---|---|
| P2–P3 | 59.5 | 9.6 | 9 | 2.38 | 0.62 | 9 | 47.6 | 21.7 | 9 | −19.6 | 5.2 | 9 | 14.1 | 4.9 | 9 | 32.1 | 5.9 | 6 | 20 | 7.8 | 6 | 1.76 | 0.53 | 6 |
| P5 | 47.1 | 7.4 | 15 | 2.99 | 0.95 | 15 | 24.5 | 10.8 | 15 | −16.9 | 4.6 | 15 | 20.1 | 4.5 | 15 | 17.7 | 5 | 7 | 8.8 | 2.1 | 7 | 2.07 | 0.71 | 7 |
| P6 | 49.0 | 9.2 | 19 | 2.99 | 0.71 | 19 | 25.1 | 19.2 | 19 | −18.0 | 6.6 | 19 | 21.0 | 3.7 | 19 | 22.7 | 6.5 | 10 | 11.7 | 3.7 | 10 | 1.97 | 0.27 | 10 |
| P7 | 46.3 | 7.8 | 35 | 2.92 | 0.64 | 35 | 23.8 | 8.6 | 35 | −17.6 | 4.1 | 35 | 22.6 | 3.7 | 35 | 17.5 | 4.7 | 13 | 8.83 | 2.4 | 13 | 2.08 | 0.71 | 13 |
| P8–P9 | 54.4 | 7.4 | 20 | 2.74 | 0.82 | 20 | 32.8 | 14.8 | 20 | −20.2 | 5.9 | 20 | 20.9 | 4.2 | 20 | 18.1 | 4.3 | 13 | 10.4 | 3.3 | 13 | 1.80 | 0.31 | 13 |
| P10–P11 | 50.6 | 11.6 | 11 | 2.51 | 0.83 | 11 | 36.3 | 21.2 | 11 | −21.4 | 7.5 | 11 | 20.7 | 4.9 | 11 | 11 | 4.5 | 10 | 11.5 | 5.3 | 10 | 1.01 | 0.30 | 10 |
| P12–P13 | 62.1 | 6.3 | 18 | 1.86 | 0.27 | 18 | 54.2 | 14.5 | 18 | −29.7 | 4.2 | 18 | 23.0 | 6.0 | 18 | 13.2 | 5 | 15 | 11.6 | 4.5 | 15 | 1.29 | 0.80 | 15 |
| P14 | 61.0 | 7.1 | 16 | 1.91 | 0.54 | 16 | 54.9 | 18.4 | 16 | −30.9 | 8.5 | 16 | 24.1 | 5.5 | 16 | 12.2 | 4 | 10 | 9.8 | 2.6 | 10 | 1.30 | 0.50 | 10 |
| P15–P16 | 60.1 | 6.5 | 30 | 1.63 | 0.49 | 30 | 56.2 | 18.3 | 30 | −39.7 | 11.5 | 30 | 26.1 | 4.0 | 30 | 9.28 | 3.2 | 26 | 9.28 | 4.2 | 26 | 1.11 | 0.44 | 26 |
| P17 | 62.1 | 8.6 | 28 | 1.61 | 0.37 | 28 | 58.2 | 16.2 | 28 | −38.8 | 8.8 | 28 | 28.1 | 5.0 | 28 | 7.13 | 2.1 | 14 | 8.28 | 2.9 | 14 | 0.92 | 0.31 | 14 |
| P18 | 55.5 | 4.7 | 21 | 1.45 | 0.21 | 21 | 50.6 | 15.6 | 21 | −41.1 | 7.3 | 21 | 28.4 | 3.2 | 21 | 6.97 | 2 | 12 | 7.59 | 2.9 | 12 | 0.99 | 0.35 | 12 |
| P19 | 58.2 | 8.8 | 33 | 1.55 | 0.38 | 33 | 54.3 | 21.1 | 33 | −41.4 | 11.4 | 33 | 27.4 | 4.6 | 33 | 8.32 | 2.6 | 19 | 9.3 | 2.6 | 19 | 0.93 | 0.31 | 19 |
| P20 | 61.6 | 8.8 | 24 | 1.47 | 0.31 | 24 | 59.3 | 21.2 | 24 | −42.6 | 8.8 | 24 | 28.0 | 5.0 | 24 | 8.15 | 2.4 | 16 | 9.21 | 3 | 16 | 0.95 | 0.33 | 16 |
| P21–P23 | 62.6 | 8.2 | 24 | 1.50 | 0.37 | 24 | 61.5 | 21.0 | 24 | −44.1 | 11.4 | 24 | 29.1 | 4.3 | 24 | 6.79 | 2.1 | 24 | 7 | 1.8 | 24 | 1.01 | 0.32 | 24 |
| P28–P29 | 62.4 | 5.3 | 12 | 1.49 | 0.29 | 12 | 63.0 | 14.3 | 12 | −43.2 | 8.7 | 12 | 28.6 | 6.2 | 12 | 6.99 | 2.3 | 10 | 6.82 | 1.4 | 10 | 1.07 | 0.43 | 10 |

Abbreviations: ISI$_{avg}$, averaged interspike interval; CV$_{ISI}$, coefficient of variation of the interspike interval; τ$_m$, membrane time constant; R$_{in}$, input resistance; C$_m$, membrane capacitance; Rebound, rebound delay; AP, action potential; AHP, after hyperpolarization; SFA, spike frequency adaptation.

sustained depolarization (*Vandecasteele et al., 2011*), we measured the gain of the spiking response (in Hz/100 pA) both at the start of the depolarizing pulse (gain at start or GS, average frequency of the first three APs) and at the end of the depolarizing pulse (gain at end or GE, average frequency of the last three APs). An SFA index was then extracted by calculating the GS/GE ratio. While GS was found to gradually decrease over the first three postnatal weeks (*Figure 3*, *Figure 7C*, *Table 1*), GE decreased mainly between P2–3 and P5 and remained stable afterwards (*Figure 3*, *Figure 7D*, *Table 1*). As a consequence of these different developmental timecourses, SFA index was found to be stable between P2–3 and P8–9 and then to drop to a steady value (*Figure 3*, *Figure 7E*, *Table 1*).

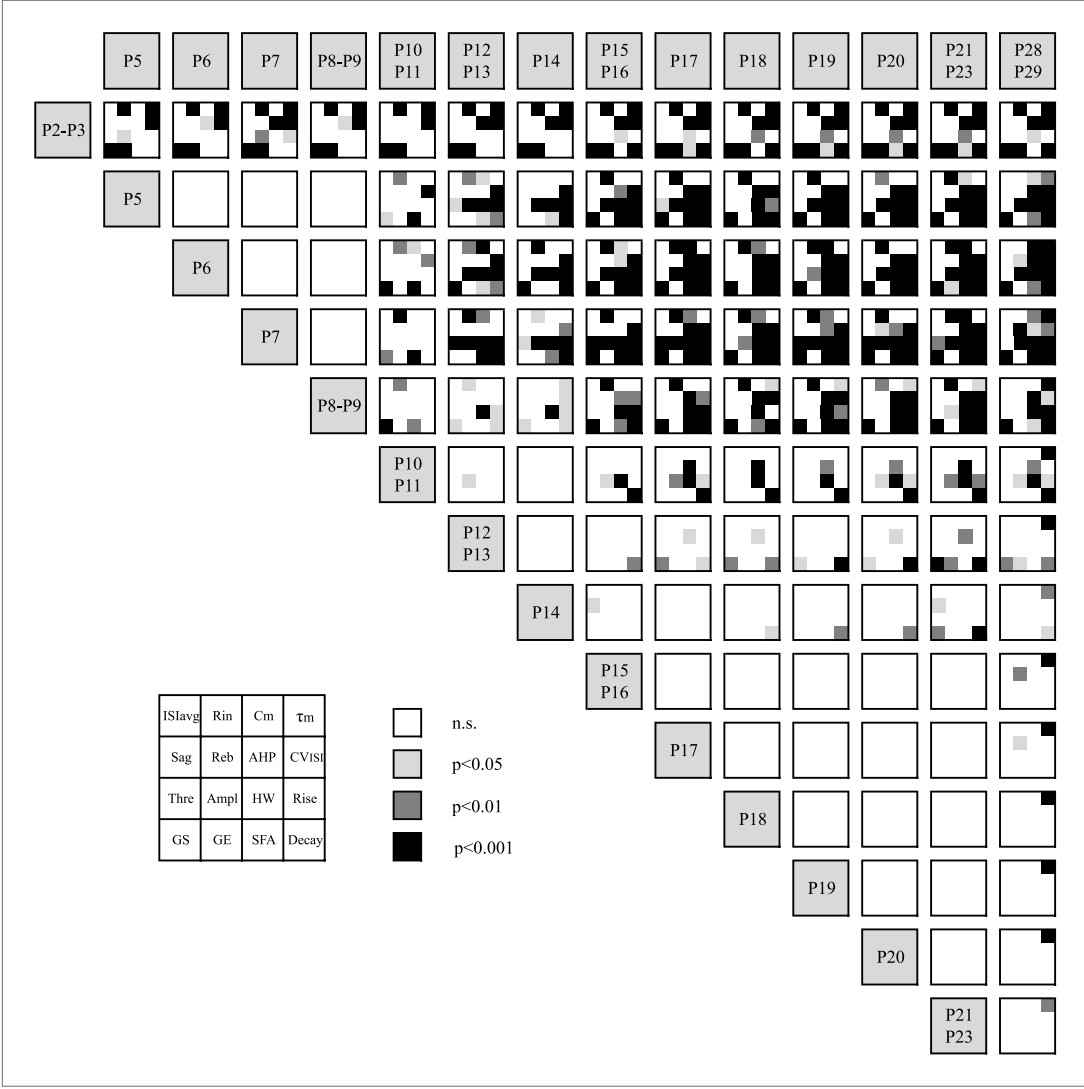

**Figure 3**. Statistical stacking table summarizing the statistical differences in 16 electrophysiological parameters across 15 developmental stages. Each major cell corresponding to the comparison between two developmental stages is subdivided in 16 sub-cells corresponding to the 16 electrophysiological parameters as depicted in the inset. The color of each sub-cell indicates the level of significance of the statistical comparison (white, non-significant; light gray, $p < 0.05$; dark gray, $p < 0.01$; black, $p < 0.001$). Statistical comparisons were performed for each electrophysiological parameter separately using a one-way ANOVA with post-hoc Tukey correction for multiple comparisons. Abbreviations: $ISI_{avg}$, averaged interspike interval; $R_{in}$, input resistance; $C_m$, membrane capacitance; $\tau_m$, membrane time constant; Reb, rebound delay; AHP, afterhyperpolarization; $CV_{ISI}$, coefficient of variation of the interspike interval; Thre, AP threshold; Ampl, AP amplitude; HW, AP half-width; Rise, AP rise slope; Decay, AP decay slope; GS, gain start; GE, gain end; SFA, spike frequency adaptation.

## Identifying the key electrophysiological transitions during postnatal development

So far, we have presented a uni-variate analysis of 16 electrophysiological properties encompassing the general electrophysiological behavior of SNc dopaminergic neurons. While this analysis is very useful to determine the developmental evolution of each distinct parameter, it fails to provide a global vision of the key moments where most significant changes in electrophysiological behavior occur: some properties such as AHP amplitude, GS or $CV_{ISI}$ display a gradual decrease over the first four postnatal weeks; some properties, such as AP amplitude, rise slope or half-width, present biphasic changes; and other properties do not present significant changes during the same developmental

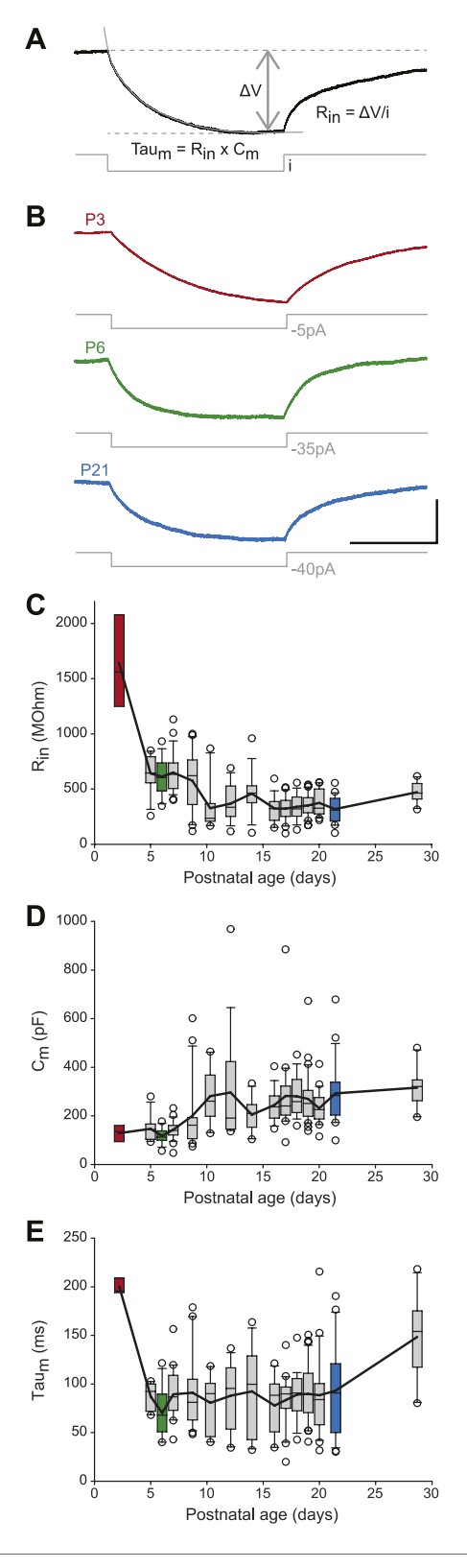

**Figure 4**. Postnatal evolution of passive properties in substantia nigra pars compacta dopaminergic neurons. (**A**), parameters extracted to characterize the passive
*Figure 4. Continued on next page*

timeframe, such as AP threshold, sag amplitude, or rebound delay.

In order to get a more global picture of the development of electrophysiological behavior, we started by representing the significant differences in each firing property between the 14 developmental stages analyzed in a 'statistical stacking table' (*Figure 3*). In this table, each major cell (plain black line surrounding) corresponds to one developmental stage, and the 16 minor cells constituting each major cell correspond to the electrophysiological properties presented in *Figures 2, 4, 5, 6 and 7*. The color coding of each minor cell then matches the level of statistical significance obtained with a one-way ANOVA comparison with post-hoc tukey correction for multiple comparisons performed between all developmental stages for each electrophysiological parameter: black corresponds to $p < 0.001$, dark gray to $p < 0.01$, and light gray to $p < 0.05$. Although this table only represents independent uni-variate analyses of the changes in each firing property, the stacking of the statistical differences provides a first visual impression of where important transitions in electrophysiological behavior occur: a first transition occurs between P2–3 and P5 while a second transition occurs between P8–9 and P12–13 (*Figure 3*). Therefore, the second transition was less clearly defined, spanning several of the developmental stages analyzed (a small number of statistical differences are still present between P10–P11 or P12–P13 and later stages). Outside of these transition phases, most electrophysiological parameters remain constant (between P5 and P8–9, then between P14 and P28). Although the stacking of uni-variate analyses gives a good indication of when major transitions are occurring, it still fails to easily and rapidly identify which properties most significantly change during the two transitions.

## Multi-variate analyses of the key electrophysiological transitions

The two major transitions identified in *Figure 3* supposedly separate three different developmental classes of electrophysiological behaviors associated with significant changes in the values of the intrinsic properties analyzed so far. In order to confirm this hypothesis and determine the key/important changes in properties involved, we performed agglomerative hierarchical clustering (AHC) analysis. We chose to include only the 8 electrophysiological parameters that can be extracted from the recordings of spontaneous activity ($ISI_{avg}$, $CV_{ISI}$, AP threshold, AP amplitude, AP

*Figure 4. Continued*

properties of dopaminergic neurons. The input resistance ($R_{in}$), membrane capacitance ($C_m$) and the membrane time constant ($Tau_m$) were calculated based on the voltage response (top, black trace) of the neuron to a small step of hyperpolarizing current (bottom, gray trace). (**B**), voltage traces (black) obtained in response to current steps (gray) of different amplitudes in P3 (top), P6 (middle) and P21 (bottom) dopaminergic neurons. (**C**), box and whisker plot representing $R_{in}$ vs postnatal age. (**D**), box and whisker plot representing $C_m$ vs postnatal age. (**E**), box and whisker plot representing $Tau_m$ vs postnatal age. Scale bars: **A** and **B**, vertical 5 mV, horizontal 250 ms. Dotted lines in **A** indicate the voltage values used to calculate ΔV. For all box and whisker plots, boxes represent the median, first and third quartile, error bars correspond to 10 and 90%, the thick line corresponds to the mean, and all outliers are represented. Colored boxes in **C**, **D** and **E** correspond to the age of the colored traces presented in **B** (red for P3, green for P6, blue for P21).

rise slope, AP decay slope, AP half-width, and AHP amplitude) and excluded the parameters that require current injection to be measured ($R_{in}$, $C_m$, $τ_m$, sag amplitude, rebound delay, GS, GE, SFA index), the goal being to determine whether phenotype can be accurately defined with minimal electrophysiological manipulation (no current injection). Using AHC with an automatic setting of the dissimilarity threshold (XLSTAT software), the 263 recorded neurons were separated into three classes of heterogeneous sizes (*Figure 8A*). The observation of the recordings corresponding to the central objects of each class reveals that bursting, irregular and regular tonic neurons were clustered in separate classes (class 1, 2 and 3, respectively; *Figure 8B*). Unsurprisingly, $CV_{ISI}$ was one of the parameters presenting the most significant differences between the central objects (*Figure 8B*) or the centroids (*Figure 8C*) of each class. Nonetheless, other parameters such as AP half-width, AP rise slope, AP decay slope, and AHP amplitude also presented notable differences between the three classes (*Figure 8B,C*). Although age was not included as one of the variables in the AHC analysis, the age of the animals corresponding to each recorded neuron was known such that we were able to determine the age of each observation (*Figure 8A*), of the central objects (see *Figure 8B*) but also the mean age for each class (*Figure 8C*). Consistent with the developmental timecourse of firing pattern depicted in *Figure 2E*, the mean ages were P3.5, P7.44, and P17.32 for class 1, 2, and 3, respectively (*Figure 8C*). Therefore, AHC analysis based solely on spontaneous activity-related electrophysiological parameters clearly discriminates between the three developmental stages (or electrophysiological behaviors) observed between P2 and P28.

Based on the parameter values of the centroids of each class, it is clear that most parameters do not follow a similar trajectory during development ($CV_{ISI}$ decreases monophasically while AP rise slope and AP amplitude present biphasic changes). Thus, we sought an analysis method that would provide a precise visualization of the developmental trajectory of the electrophysiological behavior of SNc dopaminergic neurons. By reducing a high-dimensional parameter space into two or three principal components that account for most of the variance observed in a sample, PCA allows coordinated changes in parameter values to be visualized. We ran PCA based on the covariance matrix (n) of the same observations/variables table used for the AHC analysis. The first two principal components (PC1 and PC2) accounted for ~94% of the variance of the variables (*Figure 9A*), and therefore were chosen for the two-dimensional representation of the developmental evolution of the 8 electrophysiological parameters during postnatal development. The contribution of each electrophysiological parameter to PC1 and PC2 is represented in *Figure 9A*: while $CV_{ISI}$ and AHP mainly contribute to PC1, other parameters such as AP amplitude, AP rise slope, AP decay slope, and AP half-width contribute to a similar extent to PC1 and PC2. Finally, $ISI_{avg}$ and AP threshold show minor contributions to both PCs.

Since the AHC and the PCA were run on the same observations/variables table, we first sought to determine whether PCA and AHC gave consistent results. To do so, we plotted the PC1 and PC2 factor loadings of the three classes identified by the AHC. Consistent with the AHC results, the three classes were found to cluster in distinct regions of the PC1/PC2 space (*Figure 9B*). Since AHC classes were found to be associated with specific developmental stages (*Figure 8C*) and clustered in separate regions of the PC1/PC2 space, we wondered whether PCA could give more specific insights into the developmental trajectory of the electrophysiological properties of SNc dopaminergic neurons. Although age was not included as one of the parameters of the PCA, the mean PC1 and PC2 values and the associated standard deviations were calculated for each developmental stage analyzed, and could be represented in the PC1/PC2 space (*Figure 9C*): the distribution of values demonstrated that the developmental trajectory of the electrophysiological properties of SNc dopaminergic

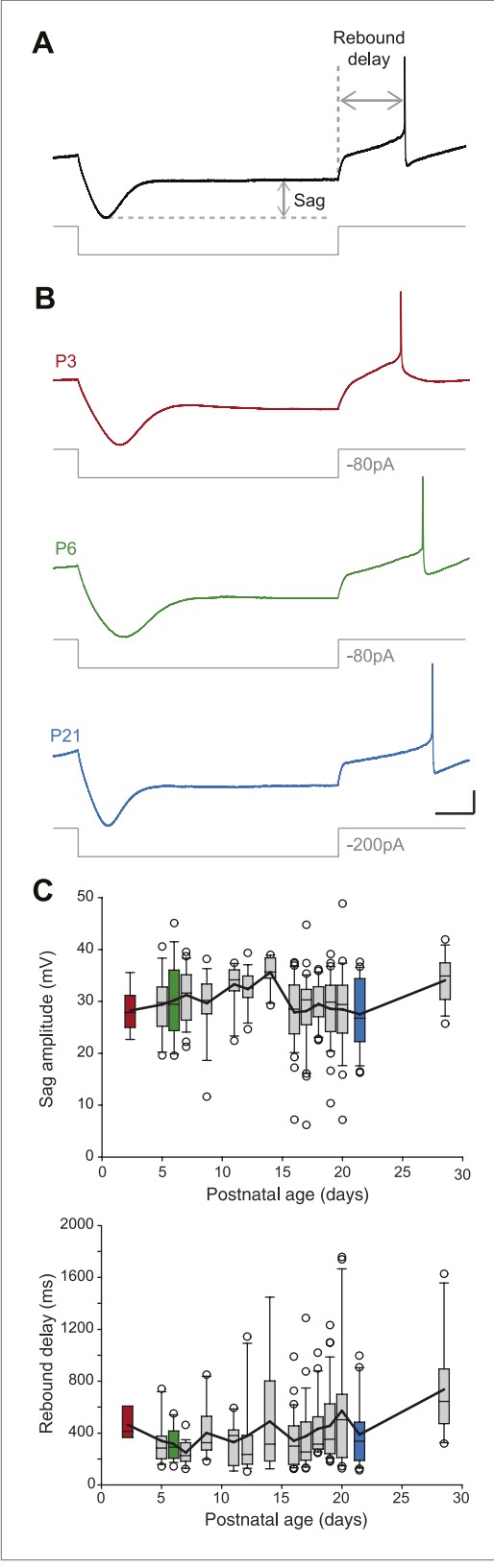

**Figure 5**. Postnatal evolution of sag and rebound delay in substantia nigra pars compacta dopaminergic neurons. (**A**), sag amplitude and delay were extracted
*Figure 5. Continued on next page*

neurons is composed of two distinct phases, a first phase where PC1 and PC2 simultaneously decrease (between P2–3 and P5–11) and a second phase where PC1 keeps on decreasing while PC2 increases (between P11 and P15) (*Figure 9C1*). Interestingly, PCA also reveals that the electrophysiological behavior of SNc dopaminergic neurons in vitro reaches a steady state by P14–15, and that no significant change is observed after this stage, although some variability in the electrophysiological properties might still be present (*Figure 9C2*). Therefore, PCA demonstrates that the electrophysiological development of SNc dopaminergic neurons is composed of two phases that involve changes in different sets of electrophysiological properties, most likely indicating that different ion currents evolve with distinct developmental timecourses.

Although averaging the data for each developmental stage was necessary to analyze the developmental trajectory (*Figure 9C*), another important aspect to investigate is the variability in electrical phenotype observed for a given developmental stage (*Marder and Goaillard, 2006*). In particular, since the mean electrical phenotype reaches a steady-state by P14–P15, we tried to determine whether this stabilization was also associated with changes in the variability of the phenotype. Since PCA involves a normalization step, standard deviations of the PCs can be used as an index of variability (using the coefficient of variation is meaningless in this case): the error bars representing the standard deviations of the PCs in *Figure 9B* already suggest that variability in phenotype is higher for immature classes (1 and 2) than for the mature class (3). In order to quantify the variability for each developmental stage, we extracted the values of the standard deviations of PC1 and PC2 (SD PC1 and SD PC2, respectively) for each developmental stage presented in *Figure 9C*, and considered that the overall variability in phenotype could be illustrated by the changes in the surface area of the ellipse defined by these two values ($A = \pi \times$ (SD PC1) $\times$ (SD PC2); *Figure 10*). This analysis showed that phenotype is more variable at early than late developmental stages. Consistent with this analysis, and since PC1 accounts for ~65% of the variance of our observations (vs ~28% for PC2), the standard deviation of PC1 was found to be significantly and negatively correlated with age ($r = 0.802$, $p < 0.001$, $n = 15$). In summary, the multi-variate analyses reveal that the electrical phenotype of SNc dopaminergic neurons follows a progressively narrowing biphasic developmental trajectory.

*Figure 5. Continued*

from the voltage response (top, black trace) of the neuron to a large hyperpolarizing current step (bottom, gray trace). (**B**), voltage recordings showing the typical sag and rebound delay of P3 (top, red), P6 (middle, green) and P21 (bottom, blue) dopaminergic neurons in response to hyperpolarizing current steps (gray traces). (**C**), top, box and whisker plot representing sag amplitude vs postnatal age. Bottom, box and whisker plot representing rebound delay vs postnatal age. Scale bars: **A** and **B**, vertical 20 mV, horizontal 150 ms. Horizontal dotted line in **A** indicates the voltage peak of the hyperpolarizing response. Vertical dotted line indicates the end of the current pulse used to calculate the delay. For all box and whisker plots, boxes represent the median, first and third quartile, error bars correspond to 10 and 90%, the thick line corresponds to the mean, and all outliers are represented. Colored boxes in **C** correspond to the age of the colored traces presented in **B** (red for P3, green for P6, blue for P21).

## Investigating the biophysical mechanisms involved in the variations of electrical phenotype

The question that arises next is whether a few specific biophysical mechanisms may explain the main developmental transitions in electrical phenotype. Although numerous voltage-dependent ion channels are involved in defining the electrical phenotype of SNc dopaminergic neurons and may be involved in these transitions, we focused on the role of three conductances: the apamin-sensitive calcium-activated potassium current, the TTX-sensitive sodium current, and the leak current.

First, we investigated the transition in electrical phenotype occurring between P2–P3 and P5–P7, which is characterized by a sudden decrease in $R_{in}$, $CV_{ISI}$, AP amplitude, and AP rise slope (*Table 1*, *Figure 3*). A steep increase in the amplitude of the AHP is also observed between these two stages (*Table 1*; *Figure 6H*). Since the apamin-sensitive calcium-activated potassium current responsible for the AHP has been linked to firing regularity (*Ping and Shepard, 1996*; *Wolfart et al., 2001*), and a sudden decrease in $R_{in}$ might explain changes in the properties of the AP (especially its amplitude), we tested whether the manipulation of these two parameters could reproduce the phenotypic transition observed between P2–P3 and P5–P7. Unfortunately, it is not possible to accurately simulate the increase in apamin-sensitive AHP occurring between P2–P3 and P5–P7 using dynamic-clamp, because of the calcium-dependence of this conductance. On the other hand, apamin very specifically blocks these conductances and simulates a decrease in AHP. Therefore, P7–P8 neurons were recorded, and apamin application (100 nM, to block 75–80% of the AHP; *Wolfart et al., 2001*) and injection of a dynamic clamp-generated negative leak conductance (average −0.55 nS, to simulate the increase in input resistance) were used in an attempt to reverse their electrophysiological phenotype to the P2–P3 stage. As expected, apamin application induced a decrease in AHP (*Figure 11A*, *Figure 11—figure supplement 1*) associated with an increase in $CV_{ISI}$, corresponding to a switch from regular to irregular firing (*Figure 11A*). Artificially increasing $R_{in}$ by the injection of a leak negative conductance amplified the apamin effect, resulting in the appearance of burst firing (*Figure 11A*), consistent with the results obtained by *Paladini et al. (1999)*. However, artificially subtracting leak before apamin application failed to induce a significant change in phenotype ($CV_{ISI}$ = 30% vs 27% before and after leak subtraction, respectively, n = 2). In order to determine whether these manipulations were able to precisely mimic the developmental transition, the PCA factor loadings corresponding to the neurons in control, apamin, and apamin—leak conditions were plotted in the PCA space (*Figure 11B*). This analysis demonstrated that apamin application and leak subtraction induced an electrophysiological regression of P8 neurons to a P2–P3 phenotype (*Figure 11B,E*), each isolated manipulation accounting for roughly 50% of the change in electrophysiological phenotype. The analysis of the changes in the individual electrophysiological properties also supported this conclusion (*Figure 11—figure supplement 1*).

We then wondered whether we could also reverse the electrical phenotype of P15 neurons to the P7 stage. The transition between P7 and P15 is mainly characterized by an increase in AHP and AP amplitude (and an increase in the AP rise and decay slopes), but no dramatic change in $R_{in}$ (*Table 1*; *Figure 3*, *Figure 5*, *Figure 6*). In order to reverse the electrophysiological phenotype of dopaminergic neurons from P14–16 to P7, we tested the effect of reducing AHP amplitude (using 5 nM apamin to block 20–25% of the AHP; *Wolfart et al., 2001*) and reducing the sodium current involved in the action potential (using 20 nM TTX). Both applications were performed simultaneously. As shown in *Figure 11C* (see also *Figure 11—figure supplement 1*), the apamin/TTX application induced an increase in $CV_{ISI}$ and a decrease in AP amplitude (and rise and decay slopes). Again, all electrophysiological parameters involved in the PCA were measured in these conditions (*Figure 11—figure supplement 1*), and we

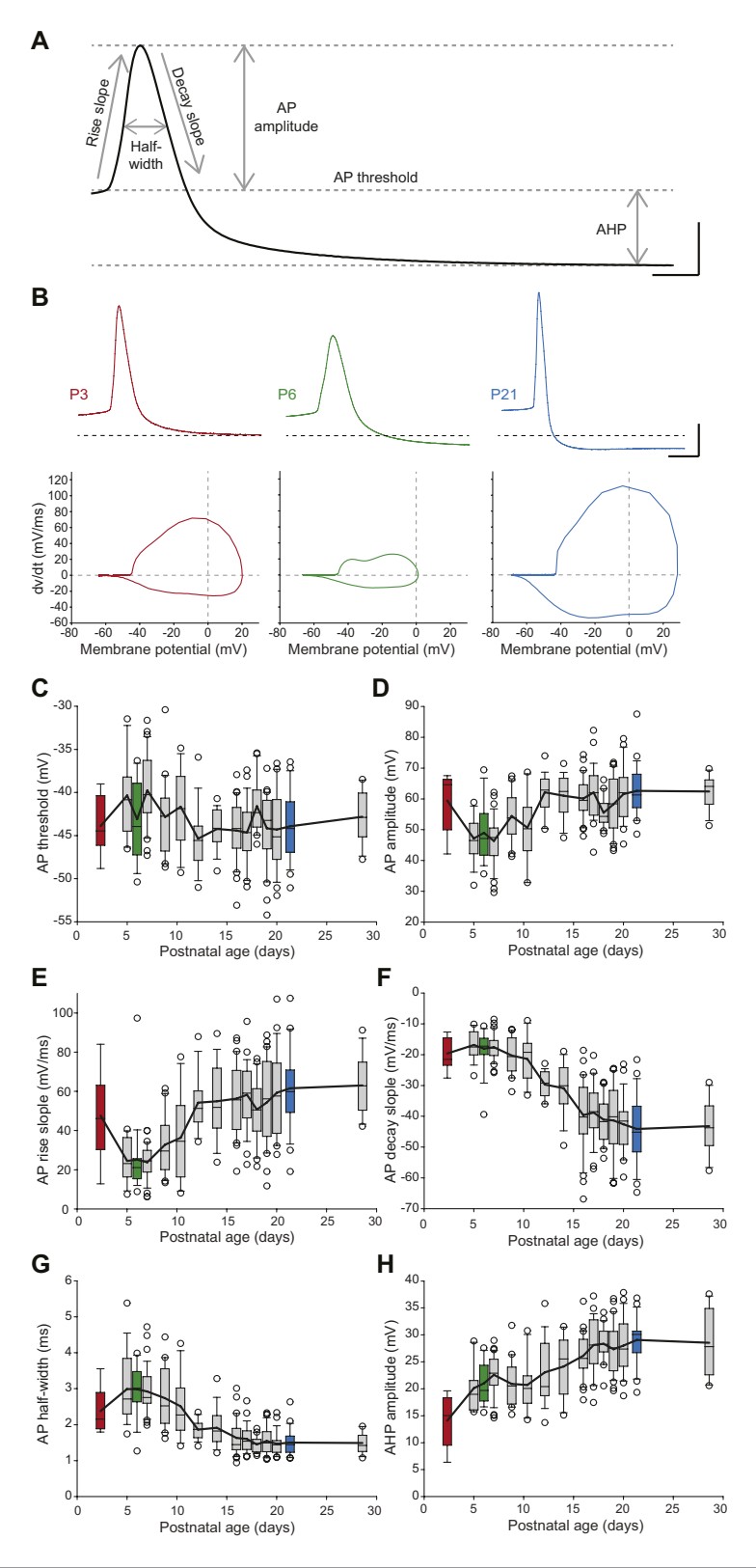

**Figure 6**. Postnatal evolution of action potential properties in substantia nigra pars compacta dopaminergic neurons. (**A**), voltage recording depicting the parameters extracted to characterize the action potential (AP) in dopaminergic neurons. Six parameters were extracted: rise slope, decay slope, half-width, amplitude, threshold

*Figure 6. Continued on next page*

*Figure 6. Continued*

and AHP amplitude. (**B**), typical APs (top traces) recorded in P3 (left, red), P6 (center, green) and P21 (right, blue) dopaminergic neurons with the corresponding phase plots representing the first time derivative of voltage vs voltage (bottom). (**C**), box and whisker plot representing AP threshold vs postnatal age. (**D**), box and whisker plot representing AP amplitude vs postnatal age. (**E**), box and whisker plot representing AP rise slope vs postnatal age. (**F**), box and whisker plot representing AP decay slope vs postnatal age. (**G**), box and whisker plot representing AP half-width vs postnatal age. (**H**), box and whisker plot representing AHP amplitude vs postnatal age. Scale bars: **A**, vertical 20 mV, horizontal 2 ms; **B**, vertical 20 mV, horizontal 5 ms. For all box and whisker plots, boxes represent the median, first and third quartile, error bars correspond to 10 and 90%, the thick line corresponds to the mean, and all outliers are represented. Colored boxes in **C**–**H** correspond to the age of the colored traces presented in **B** (red = P3, green = P6, blue = P21).

could plot the PCA factor loadings corresponding to the control and apamin/TTX-treated P14–P16 neurons in the PCA space (*Figure 11D*). This analysis demonstrated that the apamin/TTX application induced a phenotype switch quantitatively similar (and opposite in sign) to the one observed between P7 and P15 (*Figure 11D,E*).

## Discussion

In the present study, we characterized the development of electrophysiological behavior of rat SNc dopaminergic neurons in vitro over the first four postnatal weeks. We first demonstrate that SNc dopaminergic neurons switch from bursting to irregular to regular spontaneous activity pattern during the first three postnatal weeks. Using univariate analysis of 16 electrophysiological parameters (AP measurements, passive properties, excitability, etc…) and multivariate analyses (AHC and PCA) of 8 electrophysiological parameters, we demonstrate that the postnatal electrophysiological development of SNc dopaminergic neurons involves two major transitions occurring between P3 and P5 and between P9 and P11, respectively, and that mature in vitro electrophysiological behavior (regular pacemaking activity) is essentially attained by P15. Using pharmacology and dynamic-clamp, we then demonstrate that the two developmental transitions can be precisely reproduced by distinct manipulations of passive and active properties, suggesting heterogeneous timecourses of development of the underlying ion channels. This study thus proposes a comprehensive approach to (i) precisely define neuronal electrical phenotype, (ii) quantify its changes during development (or in varying conditions), and (iii) investigate the biophysical mechanisms underlying phenotypic changes.

### Postnatal evolution of spontaneous and elicited activity

So far, two main types of spontaneously occurring intrinsic patterns of activity have been described for SNc dopaminergic neurons in acute slices: a 'mature' regular tonic activity (or pacemaking) in animals above P15 (*Grace and Onn, 1989*; *Liss et al., 2001*; *Seutin et al., 2001*; *Puopolo et al., 2007*; *Guzman et al., 2009*; *Putzier et al., 2009b*; *Tateno and Robinson, 2011*; *Amendola et al., 2012*) and an 'immature' irregular pattern of activity in juvenile animals (between P6 and P15) (*Seutin et al., 1998*; *Cui et al., 2004*). Our results extend these observations by showing that an additional intrinsic pattern of activity, a bursting pattern, can be observed in acute slices from neonatal rats (before P8). As demonstrated in *Figure 2*, SNc DA neurons switch from bursting to irregular and regular activity over the first three postnatal weeks, these transitions corresponding to a non-linear monophasic drop in the coefficient of variation (inversely proportional to activity regularity) of the interspike interval. While bursting activity had been previously observed in vitro in response to glutamatergic receptor agonists (*Mereu et al., 1997*; *Johnson and Wu, 2004*) or pharmacological blockade of small-conductance calcium-activated potassium (SK) currents (*Wolfart and Roeper, 2002*; *Johnson and Wu, 2004*), we demonstrate that SNc dopaminergic neurons are purely intrinsic bursters at very early stages of development, independent of the recording condition (cell-attached, whole-cell, EGTA concentration). Interestingly, a very recent study using calcium imaging demonstrated that a significant proportion of SNc dopaminergic neurons intrinsically generate calcium spikes or calcium plateaus between E16 and P0, while this calcium activity strongly decreases by P7 (*Ferrari et al., 2012*). Since bursting behavior usually relies on the activation of calcium conductances (*Destexhe and Sejnowski, 2003*), including L-type calcium channels in SNc dopaminergic neurons (*Johnson and Wu, 2004*), our observation that most SNc dopaminergic neurons at P2–P3 are intrinsic bursters corroborates the findings of *Ferrari et al., (2012)*.

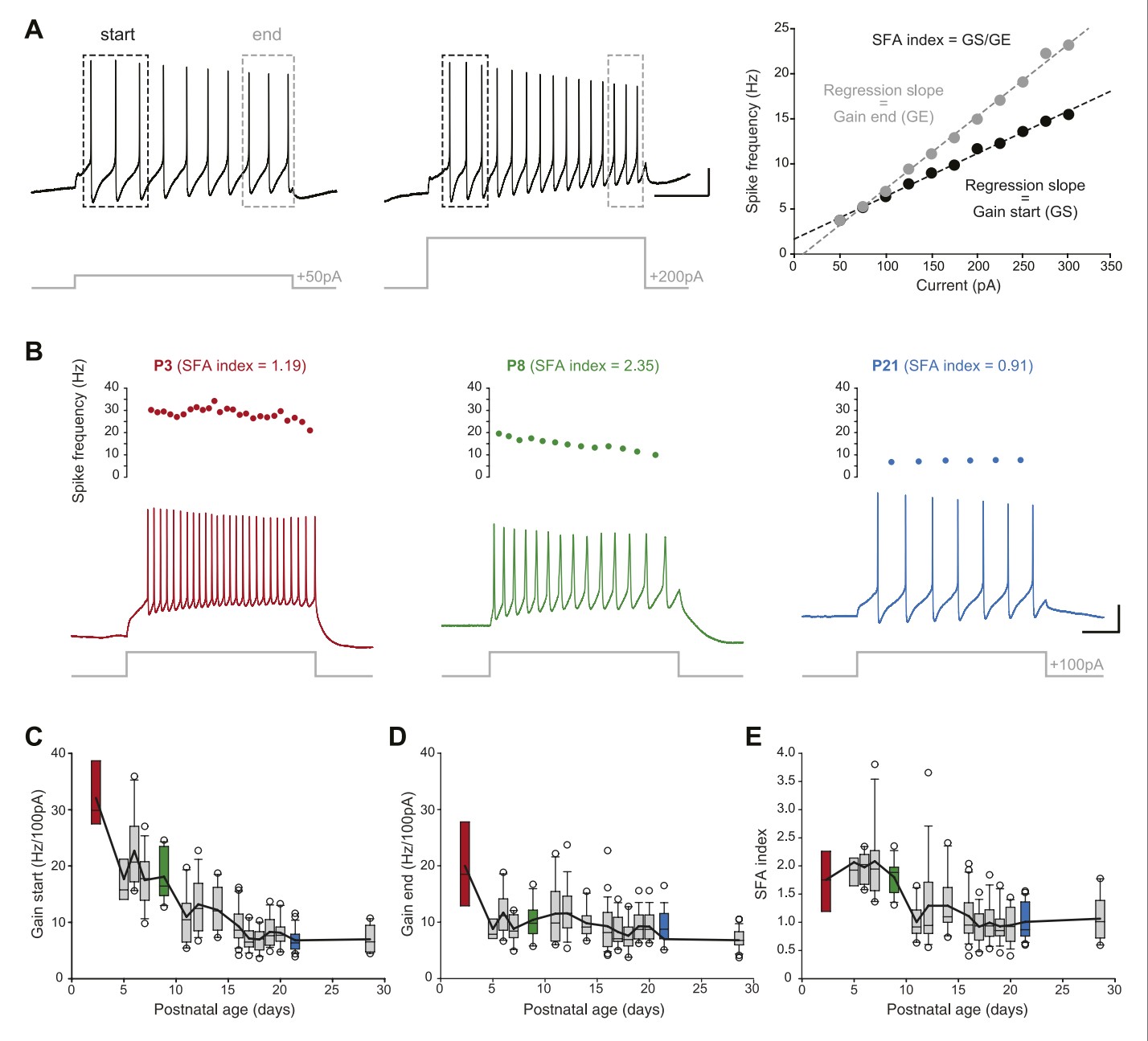

**Figure 7**. Postnatal evolution of membrane excitability in substantia nigra pars compacta dopaminergic neurons. (**A**), voltage traces depicting the parameters extracted to characterize membrane excitability. The firing frequency of the neuron was measured in response to 1 s current pulses of increasing amplitude (from 50 pA to 300 pA for the neuron shown in this panel), and the starting and ending AP frequencies were calculated for each pulse, based on the first three and last three APs, respectively (left and center panels). The gain at the start (GS) and at the end (GE) of the pulse were then extracted from the linear regression of the AP frequency vs current plot (right panel). The spike frequency adaptation (SFA) index was computed as the ratio GS/GE. (**B**), responses of P3 (left, red), P8 (center, green) and P21 (right, blue) neurons to 100 pA depolarizing pulses. Voltage recordings (bottom traces) and corresponding timecourses of instantaneous spike frequency (top plots) are shown for the three developmental stages. SFA index values calculated as presented in **A** are given for each neuron. (**C**), box and whisker plot representing GS vs postnatal age. (**D**), box and whisker plot representing GE vs postnatal age. (**E**), box and whisker plot representing SFA index vs postnatal age. Scale bars: **A**, vertical 20 mV, horizontal 250 ms. For all box and whisker plots, boxes represent the median, first and third quartile, error bars correspond to 10 and 90%, the thick line corresponds to the mean, and all outliers are represented. Colored boxes in **C**, **D** and **E** correspond to the age of the colored traces presented in B (red for P3, green for P8, blue for P21).

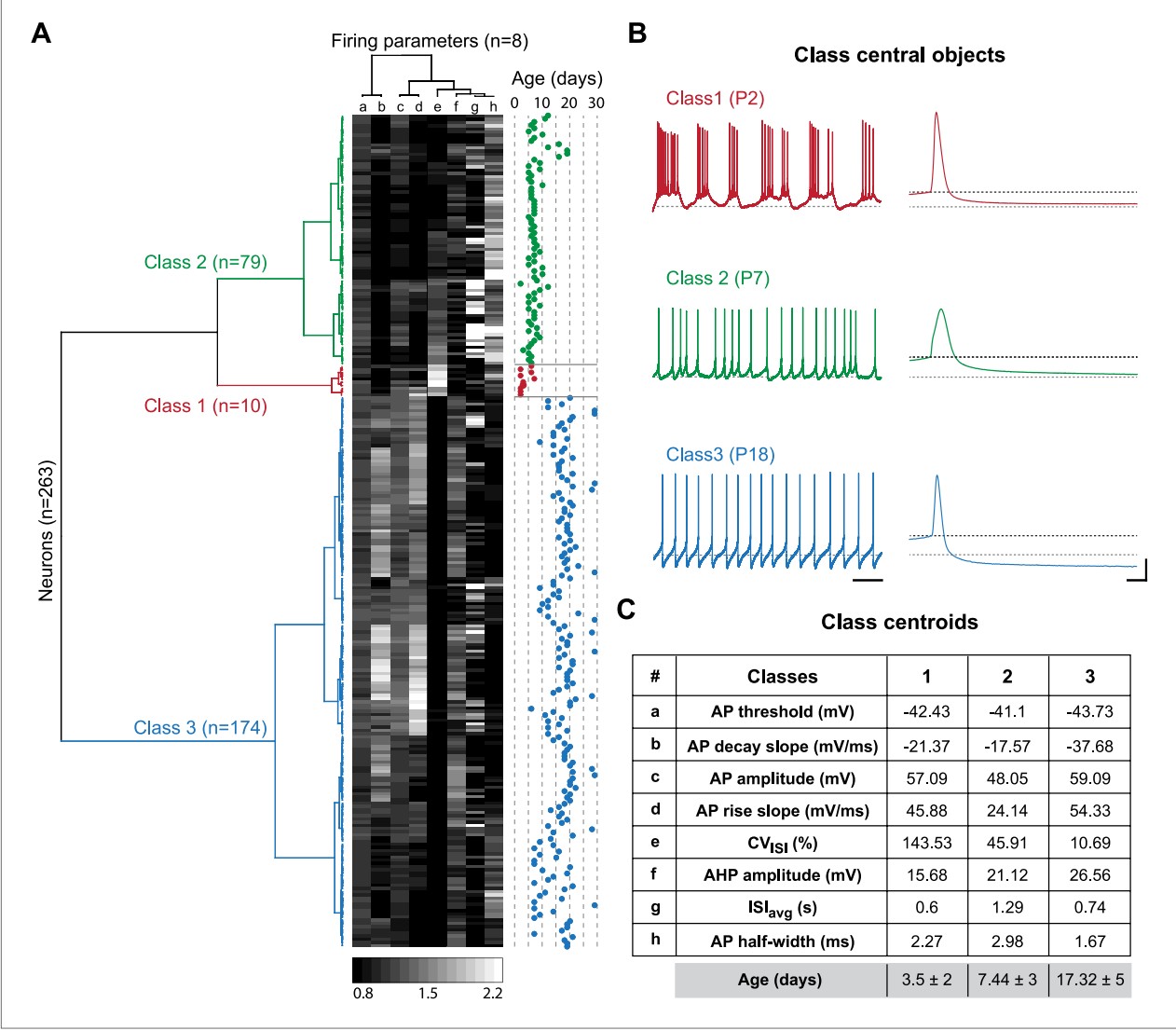

**Figure 8**. Agglomerative hierarchical clustering analysis of electrophysiological classes of substantia nigra pars compacta dopaminergic neurons during postnatal development. (**A**), dendrogram representing the agglomerative hierarchical clustering (AHC) of the 263 P2–P29 recorded neurons into three classes based on the 8 electrophysiological parameters (a to h) listed in panel **C**. The graded gray heat map represents the value of each electrophysiological parameter for each neuron normalized to the mean of the parameter for the whole population (scale bar at the bottom of the graph). The age of each neuron is plotted on the right and gives an indication of the relationship between the 3 electrophysiological classes and postnatal age. (**B**), voltage recordings corresponding to the central objects of each class showing the differences in spontaneous activity patterns and AP shape associated with each class. (**C**), table presenting the values of the 8 electrophysiological parameters for each class centroid. As an indication, the average age of each class was calculated and is given in the last row in gray (mean ± SD). Scale bars: **B**, vertical 20 mV, horizontal 2 s and 5 ms for the left and right column traces, respectively. The gray and black dotted lines in panel **B** indicate −60 mV and the AP threshold, respectively. Colors in **A** and **B** indicate the three classes (red for class 1, green for class 2, blue for class 3).

To date, however, the physiological significance of this intrinsic bursting is unknown. In several systems including the retina and the spinal cord (***Moody and Bosma, 2005***), spontaneous waves of activity early in development have been shown to participate in network development/synaptic refinement. Calcium has a central role in this process, and the patterns of activity observed in these systems (synchronized bursts of action potentials) promote intracellular calcium rises (***Moody and Bosma, 2005***). While midbrain dopaminergic neurons already project to the striatum at embryonic stage E16 and release dopamine at E18 (***Hu et al., 2004***; ***Ferrari et al., 2012***), the refinement of their projections continues at least until P0 (***Hu et al., 2004***), and it is likely that this pruning involves calcium-dependent

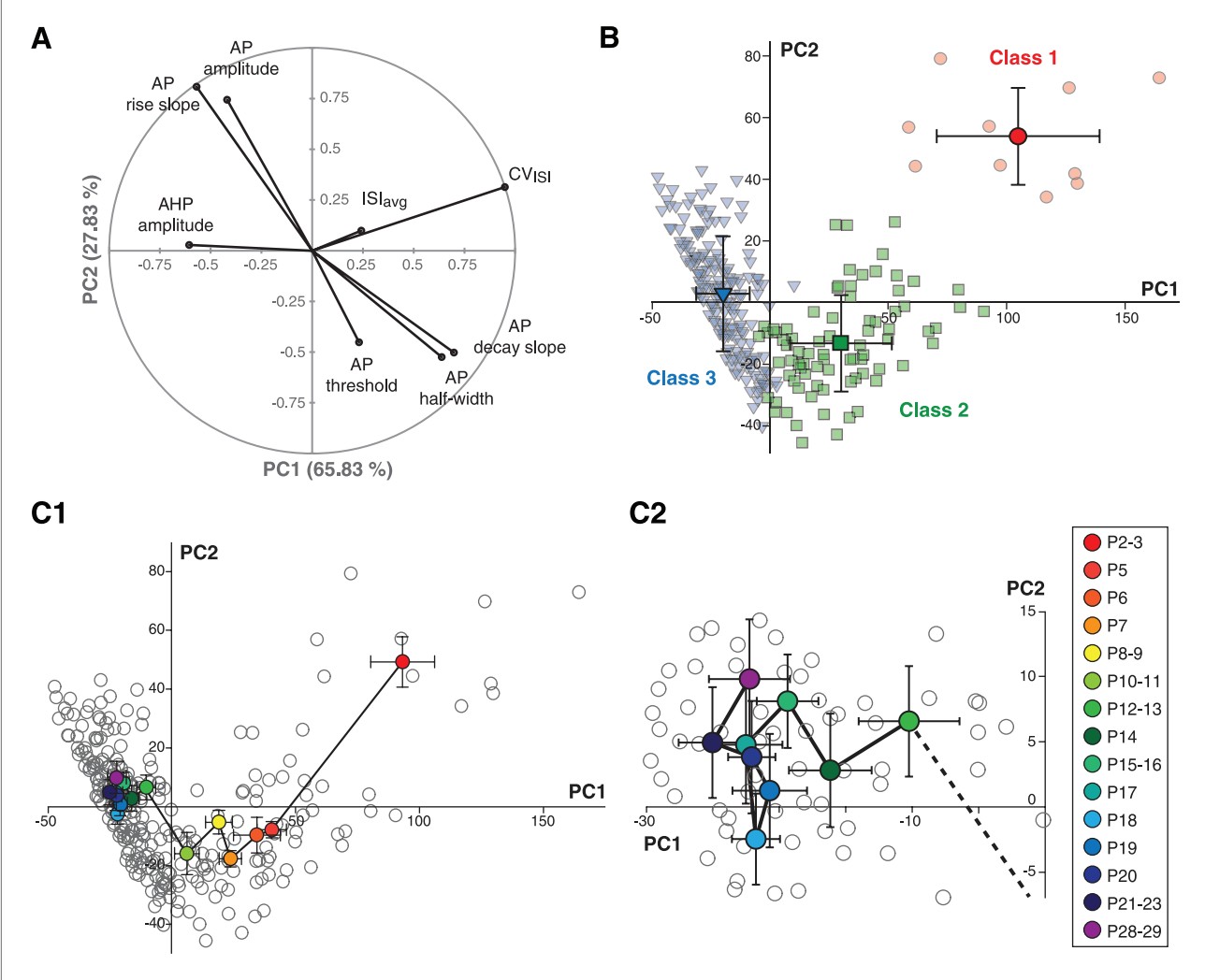

**Figure 9**. Principal component analysis of the electrophysiological behavior of substantia nigra pars compacta dopaminergic neurons during postnatal development. (**A**), polar plot representing the respective contribution of each of the 8 electrophysiological parameters to the two principal components retained from the PCA (PC1 and PC2). (**B**), scatter plot representing the factor loadings of the 263 P2–P29 neurons in the PC2 vs PC1 space. The points are color-coded as a function of the AHC class: red circles for class 1, green squares for class 2 and blue triangles for class 3. Light color symbols correspond to individual neurons while bright color symbols represent the averaged factor loadings for each class (error bars represent the SD). (**C1**), scatter plot representing the factor loadings of the 263 P2–P29 neurons (gray circles) and the averaged factor loadings for each developmental stage (colored circles). Error bars represent the standard error of the mean. (**C2**), expanded version of the scatter plot presented in **C1** centered on the later developmental stages (P12–P13 to P28–P29).

mechanisms. The intrinsic bursting observed in SNc dopaminergic neurons at early postnatal stages may promote this process.

From a biophysical point of view, regular tonic activity relies on many different voltage-dependent intrinsic conductances, including transient sodium currents (*Puopolo et al., 2007*; *Guzman et al., 2009*; *Drion et al., 2011*; *Tucker et al., 2012*), high and low voltage-activated calcium currents (*Chan et al., 2007*; *Puopolo et al., 2007*; *Guzman et al., 2009*; *Putzier et al., 2009a*), transient potassium currents (*Liss et al., 2001*; *Hahn et al., 2003*; *Putzier et al., 2009b*; *Amendola et al., 2012*), and hyperpolarization-activated currents (*Seutin et al., 2001*; *Neuhoff et al., 2002*; *Chan et al., 2007*; *Tateno and Robinson, 2011*), while the irregular pattern of activity observed in juvenile animals seems to be strongly dependent on the sporadic activation of the apamin-sensitive SK currents via the spontaneous activation of T-type calcium channels (*Seutin et al., 1998*; *Cui et al., 2004*). In fact, SK currents have been demonstrated to control the regularity of spontaneous activity in SNc dopaminergic

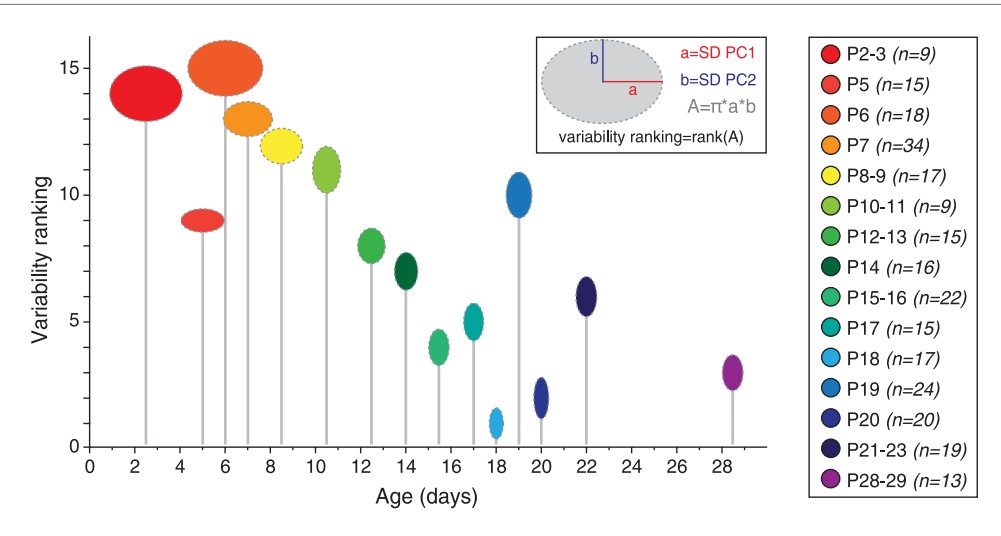

**Figure 10.** Developmental timecourse of electrical phenotype variability. Bubble plot representing the variability in electrical phenotype for each developmental stage in *Figure 9C*. The size of each ellipse is determined by the values of the standard deviations for PC1 (horizontal semi-axis, a) and PC2 (vertical semi-axis, b), as indicated in the inset. The Y-coordinate of each ellipse (variability ranking value) is then determined by the surface area A of each ellipse, which is proportional to phenotype variability as defined in the PCA space. Each ellipse is positioned along the X-axis according to the developmental stage it represents (gray drop line).

neurons from both juvenile and mature animals in vitro (*Ping and Shepard, 1996*; *Wolfart et al., 2001*). Moreover, SK current activation seems to be essential to the regular tonic pattern of activity as blockade of SK currents, under specific experimental conditions, induces a switch to a bursting pattern of activity, both in vitro (*Wolfart and Roeper, 2002*; *Johnson and Wu, 2004*) and in vivo (*Waroux et al., 2005*). To summarize these observations, although irregular juvenile activity seems to involve the activation of SK conductances (via T-type calcium currents), the 'mature' regular tonic pattern of activity also relies on the strong activation of these same conductances.

Consistent with these numerous indications of the role of SK currents in the definition of the type of activity generated by SNc dopaminergic neurons, the switch in activity pattern we observed over the first two postnatal weeks goes along with a developmental increase in the amplitude of the SK-mediated AHP (*Figure 6H*). In conclusion, our results and the published literature suggest that the gradual switch between bursting, irregular and regular intrinsic activity is at least partly due to a developmental increase in the amplitude of SK currents, or at least of the SK-mediated AHP. These results are also consistent with a recent study that demonstrated that spike frequency adaptation (SFA) in SNc dopaminergic neurons depends on SK currents, and that SFA decreases between P6 and P21 (*Vandecasteele et al., 2011*), an observation also made in the current study (*Figure 7*). Altogether, our results suggest that the developmental increase in the SK-mediated AHP over the first three postnatal weeks is essential to the maturation of SNc dopaminergic neuron spontaneous and evoked firing patterns.

The pharmacology and dynamic-clamp experiments presented in *Figure 11* confirm the central role of SK currents in defining the electrical phenotype of SNc dopaminergic neurons. Adjusting AHP amplitude to levels corresponding to specific developmental stages using apamin (see *Figure 11* and *Figure 11—figure supplement 1*) strongly contributes to reversing the overall electrical phenotype of SNc dopaminergic neurons to these stages.

## Postnatal evolution of passive and subthreshold properties

The electrophysiological measurements of passive properties (input resistance, membrane capacitance, membrane time constant) we obtained in the current study extend previous results on the morphological changes of SNc dopaminergic neurons during postnatal development: SNc dopaminergic neurons were shown to reach morphological maturity by P14, since neither soma size (*Tepper et al., 1994*) nor

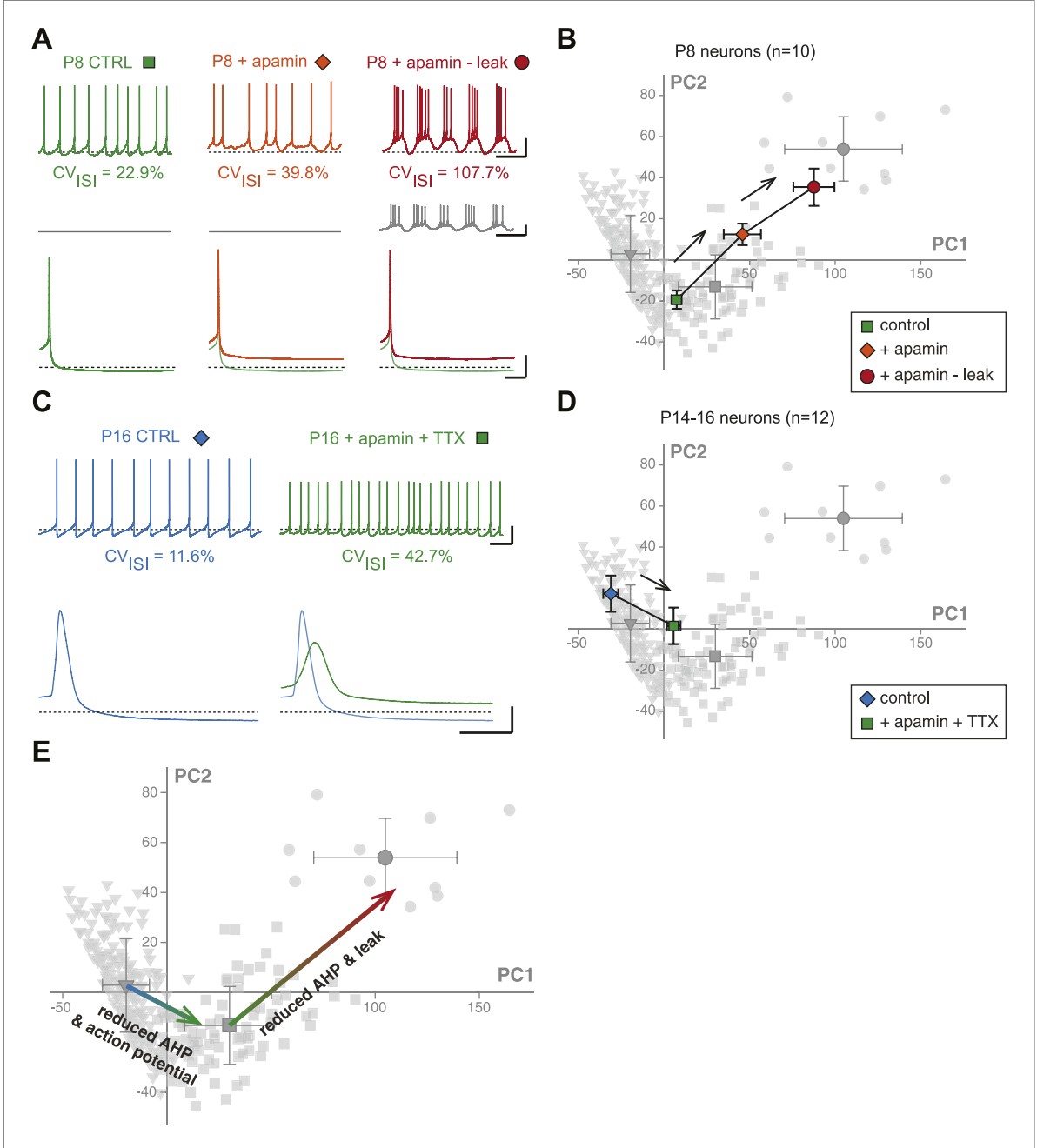

**Figure 11**. Reproducing the developmental transitions in electrical phenotype using pharmacology and dynamic clamp. (**A**), current-clamp recordings (top traces) obtained from a P8 neuron in control condition (left, green), during apamin application (center, orange), and during apamin application + negative leak injection (right, red). The middle traces (gray) correspond to the dynamic clamp-injected current while the lower traces show the changes in shape of the AHP in the different conditions. (**B**), average factor loadings obtained for 10 P8 neurons recorded in the three conditions presented in **A** and plotted in the PCA space presented in *Figure 9B*. The background gray points correspond to the three classes identified using AHC, and are plotted as a reference. (**C**), current-clamp recordings (top traces) of a P14 neuron recorded in control condition (left, blue) and in the presence of apamin and TTX (right, green). The lower traces show the changes in the AP induced by apamin + TTX. (**D**), average factor loadings obtained for 12 P14–16 neurons recorded in the two conditions presented in **C** and plotted in the PCA space presented in *Figure 9*. The background gray points correspond to the three classes identified using AHC, and are plotted as a reference. (**E**), summary of the changes in phenotype induced by the manipulations presented in **A–B** and **C–D**. The total vectors corresponding to apamin − leak (P8 neurons) and apamin + TTX (P14–16 neurons) were locked on the averages of the classes 2 and 3 of the AHC (corresponding to average ages of P7 and P17, respectively). Scale bars: **A**, vertical 20 mV (top and bottom

*Figure 11. Continued on next page*

*Figure 11. Continued*

traces) and 10 pA (middle traces), horizontal 2 s (top and middle traces) and 50 ms (bottom traces). (**C**), vertical 20 mV, horizontal 1 s and 5 ms for the top and bottom traces, respectively. The black dotted lines in panel **A** and **C** indicate −60 mV.
The following figure supplement is available for figure 11:

**Figure supplement 1**. Effect of dynamic-clamp and pharmacological manipulations on 8 individual electrophysiological properties.

cell area (*Park et al., 2000*) change significantly from P14 to adulthood. Consistent with these observations, our measurements show that membrane capacitance is fairly stable after P10–P11. Moreover, our data seem to indicate that membrane capacitance is also fairly stable, at lower values, between P2–P3 and P8–P9. Input resistance, however, shows a different developmental profile, with a very sudden drop between P2–P3 and P5, and then a gradual decrease until P10–P11. Therefore, the detailed sampling of early developmental stages seems to indicate that the morphology-related passive properties of SNc dopaminergic neurons reach maturity by the middle of the second post-natal week (P10–P11), extending previous morphological measurements (*Tepper et al., 1994*; *Park et al., 2000*).

Surprisingly, the measurements of sag ampitude and rebound delay showed that these two properties are fairly stable across the first four postnatal weeks, although $I_H$, which strongly contributes to these features (*Washio et al., 1999*; *Franz et al., 2000*; *Neuhoff et al., 2002*; *Amendola et al., 2012*), has been reported to increase over the first two postnatal weeks (*Washio et al., 1999*). Our current interpretation is that the drop in input resistance observed during the same timeframe counteracts the reported increase in $I_H$ amplitude (*Washio et al., 1999*), such that sag amplitude and rebound delay remain constant.

## Postnatal evolution of action potential properties

One of the most surprising findings of the current study is that some of the properties of the AP show a clear biphasic trajectory over the first two postnatal weeks. In particular, AP amplitude and AP rise slope decrease from P2 to P5–P7, then increase to reach a steady-state by P11–P15. So far, previously published studies on SNc dopaminergic neurons have described more classical changes in these parameters over the same time range, that is, a monophasic increase in amplitude associated with an increase in AP rise slope (or the subsequent decrease in AP half-width) (*Washio et al., 1999*; *Ferrari et al., 2012*), although fewer developmental stages and smaller samples were analyzed compared to the current study. Nevertheless, previous studies dedicated to mouse cerebral cortex and hippocampus development described the same type of monophasic increase in amplitude and acceleration of the AP (*McCormick and Prince, 1987*; *Spigelman et al., 1992*; *Picken Bahrey and Moody, 2003*; *Williams and Sutherland, 2004*). One of these studies demonstrated that it was due to an increase in the amplitude of the transient sodium current over the first two postnatal weeks (*Picken Bahrey and Moody, 2003*), consistent with the role of this current in defining the rise kinetics and the amplitude of the AP (*Bean, 2007*).

Several factors could explain the biphasic timecourse observed in the current study. First, the analysis of passive properties demonstrated that the electrotonic size of SNc dopaminergic neurons suddenly increases between P2–P3 and P5 (*Figure 4*), mainly due to a drop in input resistance. Such a drop in input resistance, in the absence of concomitant changes in sodium channel number, would cause a sudden decrease in the ability of sodium currents to depolarize the neuron, and could explain the drop in AP amplitude and AP rise slope. Other morphological changes could also explain these biphasic changes in AP properties. For instance, it is known that the axon initial segment (AIS), the main site of AP initiation, is located at highly variable distances from the soma (between 0 and 250 μm) in mature SNc dopaminergic neurons (*Hausser et al., 1995*). Recent studies have demonstrated that mild changes in the location of the AIS can significantly influence the response of the neuron to sensory inputs (*Kuba et al., 2006*, *2010*) or current injection (*Grubb and Burrone, 2010a*, *2010b*). Along the same line, and since P2–P3 SNc dopaminergic neurons have a significantly smaller size than at later stages, one could hypothesize that these changes are associated with significant displacements of the AIS that would alter AP properties. However, this hypothesis would need further investigation.

## Statistical stacking as a way to reveal the key developmental transitions

When a phenotype cannot be captured by a single characteristic and is better described by many distinct properties, as is the case for the electrophysiological phenotype of SNc dopaminergic neurons, analyzing all parameters in parallel using univariate analysis fails to provide a satisfying picture of the phenotypic variations, especially if the different parameters do not co-vary. While multivariate analyses such as PCA are designed to solve that problem by compressing the number of meaningful parameters (see next section), we propose an intermediate solution which consists of a visual stacking of the results of the uni-variate analyses performed on each parameter. Dimensional stacking entails visualizing a high-dimensional data set in a two-dimensional space and has been recently applied to analyze the electrical behavior of databases of complex realistic neuron models (*Taylor et al., 2006*). In the current study, we performed ANOVA on 16 electrophysiological parameters across 14 developmental stages, thus, to compress the results into two visible dimensions, we (i) designed a two entry-table comprising 105 cells corresponding to all stage-to-stage comparisons (P2–P3 vs P5, P2–P3 vs P6, etc) (ii) divided each of these 105 cells into 16 sub-cells corresponding to the 16 electrophysiological parameters analyzed at each developmental stage, and (iii) color-coded the sub-cells as a function of the statistical significance of the ANOVA test performed (non significant, $p < 0.05$, $p < 0.01$, $p < 0.001$). As a consequence, although *Figure 3* strictly represents the results of separate univariate analyses, it also provides a general view of the changes in the electrical phenotype of dopaminergic neurons, since each cell summarizes the differences in 16 electrophysiological parameters between two specific developmental stages: while the evolution of 1 electrophysiological parameter can be tracked through development by looking at one specific sub-cell, looking at the mosaic pattern of the large cells gives an indication of the general change in electrophysiological phenotype. This type of stacked representation is likely to be useful to anyone analyzing changes in a large number of parameters across a large number of states, and is increasingly used when analyzing the behavior of databases of models (*Taylor et al., 2006*; *Gutierrez et al., 2013*). In our case, the statistical stacking revealed two main transitions in electrical phenotype occurring between P2–P3 and P5 and between P8–P9 and P12–P13, respectively.

## Using multivariate analysis to analyze the key developmental transitions

So far, multivariate analyses, and in particular clustering analysis, have been mainly used in neurophysiology to classify and distinguish mature neuronal populations (*Ascoli et al., 2008*; *Karagiannis et al., 2009*; *McGarry et al., 2010*; *Laramee et al., 2013*; *Simonnet et al., 2013*). For instance, research dedicated to the classification of cortical interneuron subpopulations (for review see *Ascoli et al., 2008*; *Defelipe et al., 2013*) has led to the creation of an international consortium (the Petilla Interneuron Nomenclature Group). This type of classification often relies on the use of hierarchical clustering analysis or similar techniques applied to a combination of morphological (axon and dendrite length and shape…), molecular (expression of specific genes), and electrophysiological (frequency of spiking, frequency adaptation…) criteria (*Karagiannis et al., 2009*; *Battaglia et al., 2013*). Principal component analysis has also been used to analyze the diversity of cortical interneurons (*McGarry et al., 2010*) or the anatomical differences of cortical principal neurons between different cortical areas (*Laramee et al., 2013*).

In the current study, we show that AHC and PCA can also be efficiently applied to the analysis of developmental changes in electrical behavior in a given population of neurons. In particular, the biphasic developmental trajectory of the electrical behavior of SNc dopaminergic neurons over the first four postnatal weeks is captured by both types of analysis: AHC identifies three separate classes of neurons (P3, P7, and P17, respectively) while PCA identifies two main transitions (one between P3 and P5, the other between P9 and P12). We also show that both types of analysis give consistent results, as the three AHC classes cluster in separate regions of the PCA space (*Figure 9B*).

PCA reveals a very clear biphasic developmental trajectory, with (i) a concerted decrease in PC1 and PC2 between P2–P3 and P5 and (ii) a decrease in PC1 coupled with an increase in PC2 between P8–P9 and P12–P13. This biphasic trajectory may indicate that the different voltage-dependent ion channels involved in the spontaneous firing pattern of SNc dopaminergic neurons reach their mature expression pattern with heterogeneous timecourses. Moreover, PCA also clearly reveals the stationarity of the electrical phenotype after P12–P13 (*Figure 9C2*). Thus, the current study demonstrates that

AHC and PCA can be used to precisely analyze the developmental trajectory of a specific population of neurons, and could be applied in the future to examine variations in neuronal electrophysiological phenotypes (eg., in disease states) and to criteria other than electrophysiological parameters.

## From principal components back to the underlying biophysical mechanisms

Since each principal component is defined by the specific relative contributions of 8 electrophysiological parameters, PCA gives insight into which parameters may play a critical role in phenotype variation, and therefore, offers the possibility to investigate the participation of specific biophysical mechanisms in phenotypic variations. With appropriate methods (for instance specific toxins and dynamic-clamp), it is then possible to modify the activity of specific ion channels and measure their precise effect on phenotype (visualized in the PCA space). Using that type of approach, we were able to determine that the two developmental transitions can be explained by distinct combinations of changes in the activity of leak, sodium, and calcium-activated potassium currents. While the increase in calcium-activated potassium current is involved in both transitions, the first transition (P3 to P5) involves a dramatic concomittant increase in leak conductance whereas the second transition (P9 to P11) may rely on a concomittant increase in fast sodium conductance. Although we cannot conclude that these biophysical mechanisms are the only ones involved in these developmental transitions, our experiments demonstrate that manipulating these parameters quantitatively reproduces the changes in electrical phenotype observed between these stages (see *Figure 11E*).

## Variability of phenotype and variability of underlying conductances

The multi-variate analyses we performed not only revealed that the electrical phenotype of SNc dopaminergic neurons was following a biphasic trajectory, it also revealed that the variability in phenotype was decreasing across development (*Figure 10*). This observation is particularly interesting and consistent with the idea of canalization of development formulated by the English embryologist Conrad Hal Waddington (*Waddington, 1942*, *1957*; *Debat and David, 2001*). In this theory, Waddington postulated the following, referring in particular to organ development: *'In a canalised system of the kind we have been considering, trajectories starting from any point within a certain volume will converge to a single end point which is the corresponding steady-state…'* (*Waddington, 1957*). This precisely describes what is occurring with SNc dopaminergic neuron's phenotype as defined in the PCA space: the variability in phenotype (and the corresponding region of the PCA space) is much larger at early developmental stages (P2–P5) than at late developmental stages where all values cluster in a single point of space (*Figures 9C* and *10*). This suggests that neurons may be very heterogeneous early in development and then use flexible developmental trajectories to reach a highly 'attractive' mature phenotype.

What does this mean from a biophysical point of view? Previous studies have shown that mature neurons generate highly similar patterns of activity using variable solutions of underlying conductances, commonly displaying twofold to fourfold ranges of expression/amplitude of ion channels/currents (*Marder and Goaillard, 2006*; *Schulz et al., 2006*, *2007*; *Goaillard et al., 2009*; *Taylor et al., 2009*; *Amendola et al., 2012*). For instance, mature SNc dopaminergic neurons can generate regular tonic activity patterns of similar frequencies while displaying substantial differences in the amplitude and gating properties of $I_A$ and $I_H$ (*Amendola et al., 2012*). These findings may imply that immature neurons, showing heterogeneous phenotypes, display even larger variability in their underlying conductances. The developmental profiles of ion channel expression rapidly change over the first two postnatal weeks, as has been shown in many systems (*Moody and Bosma, 2005*), including the SNc dopaminergic neurons (*Washio et al., 1999*; *Dufour et al., 2014*). Slight differences in the developmental timecourses of neurons (visible only at early developmental stages) would certainly exacerbate the cell-to-cell variability in expression of ion channels already observed in mature neurons, and would reveal even larger ranges of expression of ion channels in immature neurons.

## Looking for better definitions of electrical phenotype

To define what a neuron's electrical phenotype is, one would need to know what the function of a given neuronal type is. Although certain neuronal types such as motorneurons or sensory neurons may have clearly identifiable functions, determining the information 'coded' by other neuronal types (such as the dopaminergic neurons analyzed in the current study) may be much more challenging and debatable. As long as we do not know what the function of the neuron is, we cannot determine which

of its electrophysiological properties are the most critical to its function (AP shape, firing frequency, average membrane potential?). How can we then define a neuron's electrical phenotype? In the current study, we proposed one defensible approach, which consists of measuring as many electrophysiological properties as possible in the same individual neuron, and considering that the electrical phenotype of the neuron is defined in this high-dimensional space of parameters. The use of multivariate analysis such as AHC and PCA then reduces the number of meaningful dimensions in order to obtain quantitative visualizations of phenotypic variations. This type of analysis is likely to become increasingly important as we move towards better defining neuronal types and phenotypes in an unbiased manner and understanding how these neuronal characteristics develop and are altered by perturbation and disease.

## Materials and methods

### Acute midbrain slice preparation

Acute slices were prepared from P2–P29 Wistar rats of either sex. All experiments were performed according to the European and institutional guidelines for the care and use of laboratory animals (Council Directive 86/609/EEC and French National Research Council). Rats were anesthetized with halothane (Nicholas Piramal India) and decapitated. The brain was immersed briefly in oxygenated ice-cold low-calcium artificial cerebrospinal fluid (aCSF) containing the following (in mM): 125 NaCl, 25 NaHCO$_3$, 2.5 KCl, 1.25 NaH$_2$PO$_4$, 0.5 CaCl$_2$, 4 MgCl$_2$, 25 glucose, pH 7.4, oxygenated with 95% O$_2$/5% CO$_2$ gas. The cortices were removed and then coronal midbrain slices (250 µm) were cut on a vibratome (Leica VT 1200S) in oxygenated ice-cold low calcium aCSF. Following 30–45 min incubation in 32°C oxygenated low calcium aCSF the slices were incubated for at least 30 min in oxygenated aCSF (125 NaCl, 25 NaHCO$_3$, 2.5 KCl, 1.25 NaH$_2$PO$_4$, 2 CaCl$_2$, 2 MgCl$_2$ and 25 glucose, pH 7.4, oxygenated with 95% O$_2$/5% CO$_2$ gas) at room temperature prior to electrophysiological recordings.

### Drugs

Picrotoxin (100 µM, Sigma Aldrich, St. Louis, MO) and Kynurenate (2 mM, Sigma Aldrich) were bath-applied via continuous perfusion in aCSF to block inhibitory and excitatory synaptic activity, respectively.

### Electrophysiology recordings and analysis

All recordings (315 cells from 120 rats) were performed on midbrain slices continuously superfused with oxygenated aCSF at 30–32°C. Picrotoxin and Kynurenate were systematically present for all recordings to prevent contamination of the intrinsic activity by spontaneous glutamatergic and GABAergic synaptic activity. Patch pipettes (1.8–4.5MΩ) were pulled from borosilicate glass (GC150TF-10, Harvard Apparatus, Holliston, MA) on a DMZ Universal Puller (Zeitz Instruments, Germany). Patch solution contained in mM: 20 KCl, 10 HEPES, 10 EGTA, 2 MgCl2, 2 Na-ATP and 120 K-gluconate, pH 7.4, 290–300 mOsm; except for 40 cells where a patch solution of same composition but containing only 0.5 mM EGTA was used (see *Figure 2—figure supplement 1*). Neurobiotin (0.05%; Vector Labs, Burlingame, CA) was included in the intracellular solution to allow identification of dopaminergic neurons using post-hoc tyrosine-hydroxylase immunolabeling (*Amendola et al., 2012*) (*Figure 1*). Whole-cell recordings were made from SNc dopaminergic neurons visualized using infrared differential interference contrast videomicroscopy (QImaging Retiga camera; Olympus BX51WI microscope) and identified as previously described (*Amendola et al., 2012*) (*Figure 1*). Whole-cell current-clamp recordings with an uncompensated series resistance <10 MΩ were included in the study, and the bridge was compensated. Capacitive currents and liquid junction potential (+13.2 mV) were compensated online and offset potentials were measured after removing the pipette from the neuron. Recordings with offset values above 1 mV were discarded from the analysis. Recordings were acquired at 10 kHz or 20 kHz (action potential measurements) and were filtered with a low-pass filter (Bessel characteristic 2.9 kHz cutoff frequency).

The interspike interval (ISI) and ISI coefficient of variation (CV$_{ISI}$) were calculated from a minimum of 40 s of stable current-clamp recording (with no injected current) within the first 5 min of obtaining the whole-cell configuration. Action potentials (APs) generated during this period of spontaneous activity were then averaged and the AP threshold, peak amplitude, rise slope, decay slope, afterhyperpolarization (AHP) amplitude (AP threshold to trough of the AHP), and the duration of the AP at half of the maximal height of the AP (AP half-width) were measured. AP threshold was defined as the membrane

voltage value reached when the voltage first derivative rises to 5% of its peak amplitude. Rise and decay slopes correspond to the average slopes measured between 10 and 90% of AP amplitude during the rise and decay of the AP, respectively. Hyperpolarizing 1 s current pulses were injected to elicit a hyperpolarization-induced sag with an average voltage value of −85 mV at the end of the hyperpolarizing current step in each cell (*Amendola et al., 2012*). To measure the passive membrane properties ($\tau_m$, $R_{in}$, $C_m$), 0.5–1 s negative current pulses inducing small hyperpolarizations (<10 mV) were used, ensuring that no hyperpolarization-activated voltage-gated current ($I_H$) contaminated the voltage signal. Incremental 1 s depolarizing pulses were injected to elicit trains of APs, and the frequency vs current curve was generated for each cell. Since SNc dopaminergic neurons display adaptation or facilitation of firing frequency during sustained current pulses (*Vandecasteele et al., 2011*), we determined the average gain value (Hz/100 pA) both for the first three and the last three APs generated during the pulse (see *Figure 7*). Therefore, gain at the start (GS) and gain at the end (GE) values were determined and the spike frequency adaptation index (SFA index) was defined as the ratio GS/GE. Dopaminergic neurons are unable to sustain firing for strong current injections, a phenomenon known as depolarization block (*Tucker et al., 2012*). This results in saturating frequencies of firing when injecting increasing depolarizing pulses of current. Our analysis of gain was focused on the 'non-saturating' firing behavior of dopaminergic neurons, that is, on the responses to current pulses inducing sustained firing of dopaminergic neurons. Thus, gain values could be extracted by linear regression of the frequency/current curve (see *Figure 7*).

Dynamic-clamp experiments were performed using the SM-2 software developed by Hugh Robinson (*Robinson, 2008*) (Cambridge Conductance, UK), which runs on a scriptable digital-signal-processing (DSP)-based system for dynamic conductance injection. Conductance definition for the leak subtraction was compiled and downloaded from the PC to a P-25M DSP board (Innovative Integration, Simi Valley, CA), which executes the conductance injection with a sampling rate of 40 KHz over a 2 V range with a resolution of 0.1 mV. The leak conductance was defined as having a reversal potential of −65 mV, and the absolute conductance value was adjusted in each neuron in order to reach an apparent input resistance of 1600 MOhm corresponding to the input resistance measured in P2–P3 neurons.

## Electrophysiology data acquisition and analysis

Data were acquired with a HEKA EPC 10/USB patch-clamp amplifier (HEKA Elektronik, Germany) and patchmaster software (HEKA Elektronik). Analysis was conducted using FitMaster v2x30 (HEKA Elektronik) and Igor Pro (version 6.0, WaveMetrics). Statistical analysis (performed according to data distribution) included: linear regression, unpaired *t* test, Mann Whitney, paired *t* test, one-way ANOVA and Fisher exact test for proportions, with a p value <0.05 being considered statistically significant. Statistics were performed utilizing SigmaPlot 10.0 (Jandel Scientific, UK) and Prism 6 (GraphPad Software, Inc., La Jolla, CA). Unless otherwise stated, data are presented as box and whisker plots, with the box representing the median value, the 25th and 75th percentiles, and the whiskers representing the 10th and 90th percentiles. In box and whisker plots, all outliers are represented. Agglomerative Hierarchical Clustering Analysis (AHC) and Principal Component Analysis (PCA) were performed using XLSTAT 2011 (Addinsoft, France). AHC dissimilarity level was calculated based on euclidian distance, and agglomeration was performed using Ward's method. The dissimilarity threshold used for defining the number of classes of objects was automatically set by the XLSTAT software. PCA was based on the covariance matrix (n) of the observations/variables table. Only the PC1 and PC2 (accounting for 93.76% of the total variance) were kept, and the corresponding factor loadings (F1 and F2) for each individual recording were analyzed and plotted. Experiments where the effects of TTX, apamin and leak negative conductance injection on firing were tested (*Figure 11*) were added as supplementary data in the PCA but were not included in the calculation of the principal components.

## Immunohistochemistry and analysis

Acute slices containing Neurobiotin tracer-filled cells were fixed 30 min in 4% paraformaldehyde at 4°C and immunolabelled with anti-tyrosine hydroxylase (Millipore, 1:9000 or Abcam, 1:2000) and Streptavidin Alexa Fluor 594 (Invitrogen; 1:12,000; 1.66 ng/ml) and donkey anti-sheep Alexa Fluor 488 (Invitrogen; 1:1000 2 μg/ml) as previously described (*Wolfart et al., 2001*; *Amendola et al., 2012*). All immunolabelling was viewed on a Leica TCS-SP2 confocal microscope (Leica Microsystems, Wetzlar, Germany), and images were captured using Leica LAS-AF software. Figures were prepared using

Sigma Plot, GraphPad Prism 5, Igor Pro, Adobe Photoshop and Adobe Illustrator (CS4, Adobe Systems Inc., San Jose, CA, U.S.A.) and ImageJ (MacBiophotonics, McMaster University, ON, Canada), with brightness and contrast adjustments performed consistently across the image to enhance clarity.

## Acknowledgements

This work was funded by Inserm (Avenir grant to J-MG, post-doctoral fellowship to JA), the Fyssen Foundation (J-MG), Conseil Général des Bouches du Rhône CG13 (J-MG) and the National Health and Medical Research Council of Australia (post-doctoral overseas training fellowship 544940 to AW), the Fondation pour la Recherche Medicale (post-doctoral fellowship to AW), the French Ministry of Research (doctoral fellowship to MAD), the Fondation France Parkinson (doctoral fellowship to MAD), and the French National Research Agency (ANR JCJC grant ROBUSTEX to J-MG). We would like to thank C Pauchet-Lopez and M Baudoux for technical assistance, and Dr F Tell, Dr D Debanne and Dr S Temporal for helpful discussions on the manuscript.

## Additional information

### Funding

| Funder | Grant reference number | Author |
|---|---|---|
| Institut national de la santé et de la recherche médicale | Avenir | Jean-Marc Goaillard |
| Fondation Fyssen | Subvention De Recherche | Jean-Marc Goaillard |
| National Health and Medical Research Council | Postdoctoral Overseas Training Fellowship 544940 | Adele Woodhouse |
| Fondation pour la Recherche Médicale | Postdoctoral Fellowship | Adele Woodhouse |
| Fondation France Parkinson | Doctoral Fellowship | Martial A Dufour |
| Agence Nationale de la Recherche | ANR JCJC | Jean-Marc Goaillard |
| Conseil Général des Bouches du Rhone | | Jean-Marc Goaillard |
| Institut national de la santé et de la recherche médicale | Postdoctoral Fellowship | Julien Amendola |
| Ministère de l'Education Nationale, de l'Enseignement Superieur et de la Recherche | Doctoral Fellowship | Martial A Dufour |

The funders had no role in study design, data collection and interpretation, or the decision to submit the work for publication.

### Author contributions

MAD, Acquisition of data, Analysis and interpretation of data, Drafting or revising the article; AW, J-MG, Conception and design, Acquisition of data, Analysis and interpretation of data, Drafting or revising the article; JA, Acquisition of data, Analysis and interpretation of data

### Ethics

Animal experimentation: All experiments were performed according to the European and institutional guidelines for the care and use of laboratory animals (Council Directive 86/609/EEC and French National Research Council).

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
