## [Decision Letter]

Thank you for sending your work entitled “Multi-dimensional analysis of electrical phenotype development in substantia nigra pars compacta dopaminergic neurons” for consideration at *eLife*. Your article has been favorably evaluated by Eve Marder (Senior editor) and 3 reviewers, one of whom is a member of our Board of Reviewing Editors.

The following individuals responsible for the peer review of your submission have agreed to reveal their identity: Ronald L Calabrese (Reviewing editor); Jorge Golowasch (peer reviewer). Two other reviewers remain anonymous.

The Reviewing editor and the other reviewers discussed their comments before we reached this decision, and the Reviewing editor has assembled the following comments to help you prepare a revised submission.

The authors present an electrophysiological analysis of the postnatal developmental trajectory of murine SNc neurons assayed in in vitro slice preparations. This is a careful, well-executed and well-written study of the electrophysiological development. While multiple previous studies have examined a few electrophysiological properties of this neuron type across development, the current study is distinct in providing: 1) comprehensive characterization of numerous sub- and suprathreshold properties, and 2) multivariate analyses of the developmental changes. In particular, these multivariate analyses (coupled with pharmacological and dynamic clamp manipulations) provide a novel, quantitative understanding of the electrophysiological development of the dopaminergic neurons, including generating hypotheses for how these developmental changes depend on underlying conductance densities. The writing is clear and the figures are well-structured and contain the necessary data. The conclusions are in general well justified in the data. The three reviewers, while coming at the paper from different perspectives have shared concerns that the authors should address.

Major concerns:

1) The authors should clarify the intent of the study and its major novelty. The paper reads something like a methods paper and has a very detailed Discussion of DA neuronal development but does not emphasize the general interest of the study. The reviewer reports all seek to get the authors to focus the paper and address general issues, albeit from slightly different but overlapping perspectives. For example, the paper does not address a major issue that is of much current interest when defining the electrical phenotype of neurons: animal-to-animal variation in conductance densities that nevertheless produce similar electrical phenotypes. The authors would enhance the interest in their data if they were to address this issue in Discussion. Moreover emphasizing the novelty of the approach not only obscures the values of the science but may overstate the case.

2) In the PCA and dendrogram analyses, only 8 of 16 measured parameters are considered and some of those considered maybe redundant, e.g., AP amplitude and rise-slope. Why were these 8 chosen and the rest ignored? Would the results have been different if more or less were included or if a different mix had been included? The authors need to justify their choices.

In the PCA analysis, the authors should clarify which parameters they chose to focus one for each component; presumably this was done based on weights (Figure 8) but the thresholds for considering a parameter were not clear.

3) Edited versions of the reviewer main reports are also provided to help with these revisions. They contain additional points to those discussed above and these like all minor comments should be addressed.

Reviewer #1:

1) Please identify the experimental system (mouse) in the title to meet *eLife* requirements. For example, the title might be “Multi-dimensional analysis of electrical phenotype development in substantia nigra pars compacta dopaminergic neurons of mice.”

2) Some minor suggested revisions are included in the manuscript pdf file. These need not be answered one by one but are suggested edits, in some cases of word choices.

Reviewer #2:

Goaillard and colleagues perform a comprehensive analysis of the electrophysiological development of substantia nigra pars compacta (SNc) dopaminergic neurons in acute slices. Of great interest, their results demonstrate multiple developmental stages, attributable in part to non-monophasic developmental changes in some electrophysiological properties, such as action potential height.

1) Throughout much of the manuscript, there is a strong emphasis placed on the Methods and Results providing a “new approach” to: 1) define an electrophysiological phenotype and, 2) define how the electrophysiological phenotype changes across development (or in response to experimental manipulation). While the comprehensive approach taken by the authors to provide univariate and multivariate analyses of electrophysiological differences is laudable (especially the pharmacological/dynamic clamp manipulations to explore biophysical mechanisms of phenotypic changes), the novelty of this approach is a bit overblown here. Thus, please amend or qualify statements of novelty throughout the manuscript to specifically reflect the application of these analyses to the developmental trajectory of the electrophysiological phenotype.

2) For the dendrogram analysis performed, it is unclear why only the 8 spontaneous firing properties (of the 16 total electrophysiological properties measured) were used. Are the clusters obtained robust to the inclusion of all electrophysiological properties measured? Likewise, how do the PCA results change with inclusion of all electrophysiological properties? Do the first two principal components continue to align predominantly with firing regularity (PC1) and AP shape (PC2)? It is unclear how “unbiased” the analyses are if they are performed on only a subset of electrophysiological properties.

3) In Figure 9 and in the corresponding text, changes in AHP amplitudes, leak conductance, and spike height are used to explain the shifts in PCA space upon pharmacological (and/or dynamic clamp) manipulations. However, given that: 1) multiple dimensions contribute to each principal component (cf., Figures 2 and 8) each ion channel current can contribute in complicated ways to multiple electrophysiological properties (cf., [4]), it is possible that the observed shifts in PCA space are the result of changes in electrophysiological properties other than (or in addition to) AHP amplitude, leak conductance, and spike height. To confirm and more explicitly show that these are indeed the main properties responsible for the shifts in PCA space, please provide quantitative data showing the direct effects of the pharmacological/dynamic clamp manipulations on all 8 electrophysiological properties used in the PCA analysis.

4) Throughout the manuscript, analysis is given for P2-P29 animals. However, the Methods state that P2-P28 rats were used. Clarify this.

5) Methods: explicitly define how action potential threshold, rising slope, decay slope were calculated.

6) Methods: unclear what “adjusted hyperpolarizing 1s pulses” means. Are these hyperpolarizing step current injections with amplitudes tuned to evoke an average steady-state hyperpolarization at -85 mV? If so, please clarify this in text of Methods.

7) The strong sag response of the membrane potential to hyperpolarizing current pulses (e.g., Figure 4) may have confounded measures of passive membrane properties (Figure 3). Please clarify how this was controlled for.

8) There appears to be a marked increase in membrane time constant from the third to fourth postnatal week (a ∼50% increase). This is potentially the most interesting result from the analysis of passive membrane properties and could have significant implications (e.g., on the integration of synaptic input and on the resonant frequency of the membrane in response to oscillatory drive). Please provide greater discussion of this finding in the manuscript.

9) Several measured properties (e.g., passive membrane properties, rebound delay, etc.) can be influenced by the resting/holding potential of a neuron. Given the spontaneous activity of these neurons, a resting/holding potential is not easy to determine. However, some description can be provided about the membrane potential in between spikes. For example, the minimum voltage reached between spikes or the average voltage xx ms before each spontaneous spike (e.g., see: Koyama and Appel, 2006) can be provided as a pseudo-resting membrane potential. Please provide some measure of the pseudo-resting membrane potential across development, for its own sake and in order to better interpret some of the other results. For example, do the neurons gradually depolarize across development? Such a change alone may help explain the transition from burst to regular firing, such as in thalamic relay cells.

10) Please clarify if the frequency-current (FI) analysis presented in Figure 6 takes spontaneous firing rates into account.

11) Did all DAergic neurons exhibit linear FI curves, similar to the representative neuron shown in Figure 6? Further, with the data that you have, are there other changes in the FI curve properties across development (i.e., dynamic range and zero-crossing point [if spontaneous rate is subtracted])?

12) There is a strong decrease in FI curve gain across development, despite a parallel decrease in input resistance and no developmental change in spike threshold. Can you speculate on the biophysical basis for this increase in gain?

13) The following statement is unclear: “Although it is clear, from the parameter values of the centroids of each class, that most parameters do not follow a similar evolution during development (CVISI decreases monophasically while AP rise slope and AP amplitude present biphasic changes).” Please clarify the meaning of this.

14) To investigate the biophysical basis underlying the 'class 1-to-class 2' transition and the 'class 2-to-class 3' transition, very different apamin concentrations were used (100 nM vs. 5 nM, respectively). Is this motivated by (or does this correspond to) quantitative differences in AHP amplitude changes between the two transitions?

15) The authors provide clear evidence that immature DAergic neurons burst, and that the transition from bursting to irregular and then regular firing is a critical aspect of the electrophysiological development of the neurons. However, no reason is given why the bursts early on might be important. Do these bursts coincide with a period of heightened synaptogenesis and/or plasticity in these neurons?

16) In addition to different internal EGTA concentrations, the time after going whole-cell can drastically affect firing patterns (e.g., see: Figure 1 of Liu and Shipley, 2008). How stable were bursting activity patterns? That is, did you observe washout of bursting?

Reviewer #3:

This manuscript is a great example of careful experimental design and execution, as well as presentation of the results. The figures are beautiful in their content and clarity, and the data are extremely convincing. I cannot say the same about the Title, which I find boring and unexciting. My enthusiasm is somewhat deflated by the authors' general presentation of the work, however. As it reads, I see a methods paper, while the conceptual novelty in it is underemphasized: you have applied these sophisticated analysis methods to a high-dimensional characterization of neuronal activity, and with this you have established 3 basic post-natal developmental stages. As the Discussion states, these were basically known, but you have refined the transition points and the features that best characterize these states. From these, and here is the most novel part from the biological point of view perhaps, 3 ionic currents have emerged as the best candidates to explain these transitions. I think that this is important work, but as I said, this should be portrayed in a broader light. The prediction from this analysis that multiple currents are involved in this transition is important but relegated to the end of the Abstract and as a secondary point (“This approach also predicts...”) and discussion. Previous work from the senior author suggesting the importance of recording many parameters from each neuron or preparation given the large variability observed in each parameter is relevant here but not mentioned. Isn't that the context where this work comes from? Also, I assume that you would predict that if I recorded from 300 adult basal forebrain cells, or leech interneurons (or any preparation), I should be able to extract the existence of subtypes that may otherwise not be known, or are only assumed to exist based on less rigorous protocols. It would be important to indicate this explicitly.

1) The first 2 sentences (ending in (26)) are rather trivial and summarized well enough in the following sentence.

2) The broad importance of this work should be expanded upon in the first paragraph of the Introduction.

3) Description of activity types is a bit ambiguous and not consistently used: Regular tonic and pacemaker are the same thing. Irregular is irregular spiking. All of these are spontaneous. If I am right, I suggest you stick with one term throughout.

4) What is the relationship between DA release and activation of IEGs?

5) What does “strongly evolves” mean?

6) Throughout the word “evolution” is used when it seem that “development” would be more appropriate.

7) What is meant by “intrinsic electrophysiological maturity”?

8) What is the statistical test used here?

9) Speaking of conductance density is reduces clarity (and “conductances' density per surface area” is redundant). I think that it is clearer to say that capacitance and leak increase, which results in a constant tau_m.

10) Why did you choose to only use those 8 parameters and not more? Also, as you data in Figure 5 show, I would argue that the AP amplitude and AP rise slope are tightly linked (dV/dt and AP amp are both dependent on I_Na). Likewise with AP half-width and AP decay slope (Figure 5 are nearly identical). Would your conclusions still hold if two parameters (one from each of these pairs of related parameters) were removed from the analyzes? I should think not since they seem to be redundant.

11) CV_ISI are said to be different and Figure 7 is referred to. I see a difference but this is not with quantitative certainty.

12) To speak of “anti-conductance” sounds very strange. It is simply a negative conductance and thus used to subtract leak.

13) 5nM apamin is used while for P7-P8 cells then100nM was used. Is this correct?

14) Table 2 and elsewhere. Isn't your statistical stacking method similar or equivalent to Taylor's clutter-based dimension reordering ? (59)

15) Table 2. I believe that using Rin, Cm and Tau_m in the same statistical comparison is not valid since Tau_m depends on the other two parameters.

---

## [Author Response]

*1) The authors should clarify the intent of the study and its major novelty. The paper reads something like a methods paper and has a very detailed Discussion of DA neuronal development but does not emphasize the general interest of the study. The reviewer reports all seek to get the authors to focus the paper and address general issues, albeit from slightly different but overlapping perspectives. For example, the paper does not address a major issue that is of much current interest when defining the electrical phenotype of neurons: animal-to-animal variation in conductance densities that nevertheless produce similar electrical phenotypes. The authors would enhance the interest in their data if they were to address this issue in Discussion. Moreover emphasizing the novelty of the approach not only obscures the values of the science but may overstate the case*.

We agree with Reviewer 1 that the paper was sort of mixing a specialist and a method perspective, and that general issues were not properly addressed in the manuscript, especially in the Discussion. We have modified the manuscript to take these comments into account. We have added a short section in the Results investigating the evolution of phenotypic variability during development (including a new Figure, Figure 9), and addressed this question in the Discussion. We also shortened the parts of the Discussion that relates to very specific points about SNc dopaminergic neurons, and tried to address more general issues relevant to a broader audience. We also modified the manuscript to make sure that the “novelty” of the approach was not overstated.

*2) In the PCA and dendrogram analyses, only 8 of 16 measured parameters are considered and some of those considered maybe redundant, e.g., AP amplitude and rise-slope. Why were these 8 chosen and the rest ignored? Would the results have been different if more or less were included or if a different mix had been included? The authors need to justify their choices*.

*In the PCA analysis, the authors should clarify which parameters they chose to focus one for each component; presumably this was done based on weights (*Figure 8*) but the thresholds for considering a parameter were not clear*.

Concerning the 8 parameters chosen for the PCA, the choice was made based on the recording condition needed to measure these 8 parameters: we specifically selected the 8 parameters that can be extracted from recordings of spontaneous activity, which do not involve any current injection. The basic question was: is it possible to accurately define electrical phenotype based on non‐manipulated recordings of spontaneous activity? This explains why these 8 parameters were chosen and why the 8 others were rejected from this analysis (sag and rebound ^rely on^ large hyperpolarizing current injections, ^R^in^, C^m ^and Tau^m ^rely on^ small hyperpolarizing current steps, while gain_start, gain_end and SFA index rely on depolarizing current injection). Although this could have been done, we did not try to optimize the number of parameters used for the PCA (by removing partially redundant properties, as suggested by Reviewer 3), but instead chose to determine which minimal (and therefore optimal) recording conditions were necessary to accurately define electrical phenotype. The results of the PCA and clustering analyses based on 8 spontaneous activity-derived parameters suggest that no current injection or other experimental manipulation of activity is needed to properly determine the phenotypic trajectory of dopaminergic cells. The thresholds based on the contribution of each property to each principal component were not considered to select the variables to be used for the PCA. We have modified the sentence introducing the multi‐variate analyses to better explain the rationale for using only these 8 specific parameters in the clustering and PCA:

“We chose to include only the 8 electrophysiological parameters that can be extracted ^from the^ recordings ^of^ spontaneous activity ^(ISI^avg^, CV^ISI^, AP^ threshold, ^AP^ amplitude, AP rise slope, AP decay slope, AP half-width and AHP amplitude), and excluded ^the^ parameters ^that^ require current injection ^to be^ measured ^(R^in^, C^m^, τ^m^,^ sag amplitude, rebound delay, GS, GE, SFA index), the goal being to determine whether phenotype can be accurately defined with minimal electrophysiological manipulation (no current injection).”

Reviewer #1:

*1) Please identify the experimental system (mouse) in the title to meet eLife requirements. For example, the title might be “Multi-dimensional analysis of electrical phenotype development in substantia nigra pars compacta dopaminergic neurons of mice*.*”*

To answer the concern raised by Reviewer 3 and the general comment about the methodological focus of the paper, we changed the title, which now underlines the main findings of the study, not the methods. Experiments were performed in rats, and the species now appears in the title as follows:

“Non-linear developmental trajectory of electrical phenotype in rat substantia nigra pars compacta dopaminergic neurons”

*2) Some minor suggested revisions are included in the manuscript pdf file. These need not be answered one by one but are suggested edits, in some cases of word choices*.

All minor revisions suggested in the manuscript pdf file have been altered in the manuscript as suggested.

In the manuscript pdf file, Reviewer 1 was asking why P30 animals were absent from the analysis (since we recorded from P28 and P29 animals). In fact, in the first run of experiments, we recorded only until P21, and ran the clustering and PCA analysis based on these data. These analyses already revealed that electrophysiological maturity was reached by P14 and that electrophysiological phenotype was stable from then on. However, since P21 animals cannot be considered as truly mature, and in order to get ahead of potential criticisms on that approximation, we decided to record at significant later stages, and extended our analysis to animals one week older (P28-P29). The analysis of P28 and P29 confirmed that electrophysiological phenotype does not significantly change after P14, including after P21 (see Tables 1 & 2 and Figures 7 and 8). This result explains why we did not feel necessary to also analyze P30 animals.

As suggested by Reviewer 1, we have modified Figure 6, which now includes color traces presenting the typical excitability response of dopaminergic neurons at specific developmental stages.

Reviewer #2:

*Goaillard and colleagues perform a comprehensive analysis of the electrophysiological development of substantia nigra pars compacta (SNc) dopaminergic neurons in acute slices. Of great interest, their results demonstrate multiple developmental stages, attributable in part to non-monophasic developmental changes in some electrophysiological properties, such as action potential height*.

*1) Throughout much of the manuscript, there is a strong emphasis placed on the Methods and Results providing a “new approach” to: 1) define an electrophysiological phenotype and, 2) define how the electrophysiological phenotype changes across development (or in response to experimental manipulation). While the comprehensive approach taken by the authors to provide univariate and multivariate analyses of electrophysiological differences is laudable (especially the pharmacological/dynamic clamp manipulations to explore biophysical mechanisms of phenotypic changes), the novelty of this approach is a bit overblown here. Thus, please amend or qualify statements of novelty throughout the manuscript to specifically reflect the application of these analyses to the developmental trajectory of the electrophysiological phenotype*.

We agree with Reviewer 2 and have amended the manuscript to restrict our statements to applying multi-variate analyses to developmental trajectory.

*2) For the dendrogram analysis performed, it is unclear why only the 8 spontaneous firing properties (of the 16 total electrophysiological properties measured) were used. Are the clusters obtained robust to the inclusion of all electrophysiological properties measured? Likewise, how do the PCA results change with inclusion of all electrophysiological properties? Do the first two principal components continue to align predominantly with firing regularity (PC1) and AP shape (PC2)? It is unclear how “unbiased” the analyses are if they are performed on only a subset of electrophysiological properties*.

As explained in the response to the major concern 2 of Reviewer 1, we specifically selected the 8 parameters that can be extracted from recordings of spontaneous activity, which do not involve any current injection. The basic question was: is it possible to accurately define electrical phenotype based on non-manipulated recordings of spontaneous activity? This explains why these 8 parameters were chosen and why the 8 others were rejected from this analysis.

Although this could have been done, we did not try to optimize the number of parameters used for the PCA (by removing partially redundant properties, as suggested by Reviewer 3), but instead chose to determine which minimal (and therefore optimal) recording conditions were necessary to accurately define electrical phenotype. As noted above we have modified the sentence introducing the multi-variate analyses to better explain the rationale for using only these 8 specific parameters in the clustering and PCA:

“We chose to include only the 8 electrophysiological parameters that can be extracted from the recordings of spontaneous activity (ISI_avg_, CV_ISI_, AP threshold, AP amplitude, AP rise slope, AP decay slope, AP half-width and AHP amplitude), and excluded the parameters that require current injection to be measured (R_in_, C_m_, τ_m_, sag amplitude, rebound delay, GS, GE, SFA index), the goal being to determine whether phenotype can be accurately defined with minimal electrophysiological manipulation (no current injection).”

Furthermore, including all 16 electrophysiological properties would have been problematic, because all properties were not measured in all neurons (some recordings were gathered from experiments that did not include the current injection protocols for passive properties, rebound and/or gain measurements, which were added in later experiments). For the reliability of the results, we did not include observations with missing data (neurons with missing properties) in our multi-variate analyses. In fact, extending the multi-variate analysis to all 16 properties would significantly reduce the sample size (from 263 neurons to less than 90). This is only a technical limitation, but an accurate definition of developmental trajectory required a significant sample at each developmental stage. Nevertheless, we recently ran the clustering analysis using 14 of the electrophysiological parameters (all but sag and rebound), which gave a population of 117 neurons comprising neurons of all ages (P2 to P29). In these conditions, 3 classes came out of the hierarchical clustering (same number as in Figure 7), with properties very similar to those described in Figure 7. As an indication, the mean ages for these classes were 2.25 (n=4), 9.3 (n=38) and 18 (n=75), respectively, very close to the ages obtained for the classes described in Figure 7 (3.5, 7.5 and 17.5). These 3 classes were also associated with contrasting CV values (115, 32 and 15%, respectively). However, the limited size of the sample (117 neurons compared to 263 in Figure 7) would compromise a detailed study of the developmental trajectory as depicted in Figure 8.

*3) In*
Figure 9
*and in the corresponding text, changes in AHP amplitudes, leak conductance, and spike height are used to explain the shifts in PCA space upon pharmacological (and/or dynamic clamp) manipulations. However, given that: 1) multiple dimensions contribute to each principal component (cf.,*
Figures 2 and 8*) each ion channel current can contribute in complicated ways to multiple electrophysiological properties (cf.,*
[4]*), it is possible that the observed shifts in PCA space are the result of changes in electrophysiological properties other than (or in addition to) AHP amplitude, leak conductance, and spike height. To confirm and more explicitly show that these are indeed the main properties responsible for the shifts in PCA space, please provide quantitative data showing the direct effects of the pharmacological/dynamic clamp manipulations on all 8 electrophysiological properties used in the PCA analysis*.

We have added a new figure (Figure 11—figure supplement 1) summarizing the changes in all 8 electrophysiological properties involved in the PCA, and have modified the Results section related to Figure 11 accordingly. As shown in this new figure, individual parameters are significantly changing in a way that is consistent with the change in the overall phenotype described in Figure 10. Following apamin application and leak subtraction, the main factors modified are CV_ISI_, AHP amplitude, AP half-width, rise and decay slopes. Following apamin and TTX application, ISI_avg_, CV_ISI_, AHP amplitude, AP threshold, amplitude and rise slope are modified.

*4) Throughout the manuscript, analysis is given for P2-P29 animals. However, the Methods state that P2-P28 rats were used. Clarify this*.

P2-P29 animals were indeed used in the study. The typographical error in the Methods section has been corrected.

*5) Methods: explicitly define how action potential threshold, rising slope, decay slope were calculated*.

Explanations have been added to the Methods section: “AP threshold was defined as the membrane voltage value reached when the voltage first derivative rises to 5% of its peak amplitude. Rise and decay slopes correspond to the average slopes measured between 10 and 90% of AP amplitude during the rise and decay of the AP, respectively.”

*6) Methods: unclear what “adjusted hyperpolarizing 1s pulses” means. Are these hyperpolarizing step current injections with amplitudes tuned to evoke an average steady-state hyperpolarization at -85 mV? If so, please clarify this in text of Methods*.

As understood by Reviewer 2, these are steps adjusted to reach a steady-state hyperpolarization of -85 mV. Since this explanation was already given, we removed the term “adjusted” to clarify the explanation.

*7) The strong sag response of the membrane potential to hyperpolarizing current pulses (e.g.,*
Figure 4*) may have confounded measures of passive membrane properties (*Figure 3*). Please clarify how this was controlled for*.

This would be true if the hyperpolarizing current pulses used to measure passive properties were large enough to induce IH activation. To measure passive properties, we used 0.5-1 s negative current pulses inducing <5mV hyperpolarizations, in order to avoid any activation of I_H_. However, we forgot to provide this information in the manuscript. A sentence has now been added to the methods section explaining which type of protocol was used to measure passive properties:

“To measure the passive membrane properties (τ_m_, R_in_, C_m_), 0.5-1s negative current pulses inducing small hyperpolarizations (<10mV) were used, ensuring that no hyperpolarization-activated voltage-gated current (I_H_) contaminated the voltage signal.”

*8) There appears to be a marked increase in membrane time constant from the third to fourth postnatal week (a ∼50% increase). This is potentially the most interesting result from the analysis of passive membrane properties and could have significant implications (e.g., on the integration of synaptic input and on the resonant frequency of the membrane in response to oscillatory drive). Please provide greater discussion of this finding in the manuscript*.

We are not exactly sure how to interpret this result, and this explains why it was not particularly discussed. Neither the capacitance nor the input resistance shows significant changes between these two stages, but their slight concomitant increases lead to a significant change in time constant. Although, as Reviewer 2 suggests, changes in membrane time constant can have an influence on the integration of synaptic inputs in specific time windows (resonance), the fact that the increase in time constant is not associated with significant changes in Rin and Cm led us to regard this result with caution. However, if requested we can add the points above in relation to the changes in membrane time constant to the Discussion.

*9) Several measured properties (e.g., passive membrane properties, rebound delay, etc.) can be influenced by the resting/holding potential of a neuron. Given the spontaneous activity of these neurons, a resting/holding potential is not easy to determine. However, some description can be provided about the membrane potential in between spikes. For example, the minimum voltage reached between spikes or the average voltage xx ms before each spontaneous spike (e.g., see:*
*Koyama and Appel, 2006**) can be provided as a pseudo-resting membrane potential. Please provide some measure of the pseudo-resting membrane potential across development, for its own sake and in order to better interpret some of the other results. For example, do the neurons gradually depolarize across development? Such a change alone may help explain the transition from burst to regular firing, such as in thalamic relay cells*.

We have not analyzed the pseudo-resting membrane potential, but we have analyzed other parameters such as the minimum voltage reached between spikes (trough of the oscillations). Although spike threshold is fairly stable over the first four postnatal weeks (Figure 5), the trough voltage clearly hyperpolarizes. This is consistent with the progressive increase in AHP amplitude observed during development (since AHP amplitude represents the difference between spike threshold and trough voltage, see Figure 5). Our conclusion is that, if anything, dopaminergic neurons hyperpolarize during development (or at least their minimum voltage does), due to the increasing participation of calcium-activated potassium currents in the activity pattern of the neurons. This is consistent with the main conclusions of the study that the apamin-sensitive AHP is a key player in the developmental changes of firing patterns observed in dopaminergic neurons.

*10) Please clarify if the frequency-current (FI) analysis presented in*
Figure 6
*takes spontaneous firing rates into account*.

It is unclear what “taking spontaneous rate into account” would mean, but no DC current was injected into the neurons outside of the depolarizing steps used to measure the frequency-current response. Although one may think that the level of spontaneous activity would influence the FI response, we found no correlation between spontaneous firing frequency and the gain measurement, suggesting that these two phenomena rely on different mechanisms.

*11) Did all DAergic neurons exhibit linear FI curves, similar to the representative neuron shown in*
Figure 6*? Further, with the data that you have, are there other changes in the FI curve properties across development (i.e., dynamic range and zero-crossing point [if spontaneous rate is subtracted])?*

Reviewer 2 mentions an important point, which we overlooked in the manuscript. It is known that dopaminergic neurons have a clearly saturating FI curve, since most of them undergo “depolarizing block” for strong current injection (see Tucker et al., J. Neurosci, 2012). Here we purposefully analyzed only the non-saturating part of the FI curve, most often stopping current steps before the neurons go into block. We have added a paragraph in the Methods section to make this clear:

“Dopaminergic neurons are unable to sustain firing for strong current injections, a phenomenon known as depolarization block (Tucker et al., J. Neurosci, 2012). This results in saturating frequencies of firing when injecting increasing depolarizing pulses of current. Our analysis of gain was focused on the “non-saturating” firing behavior of dopaminergic neurons, i.e. on the responses to current pulses inducing sustained firing of dopaminergic neurons. Thus, gain values could be extracted by linear regression of the frequency/current curve (see Figure 8).”

As mentioned in the response to comment 10, since spontaneous activity was not prevented by negative current injection, it was difficult to determine the zero- crossing point.

*12) There is a strong decrease in FI curve gain across development, despite a parallel decrease in input resistance and no developmental change in spike threshold*. *Can you speculate on the biophysical basis for this increase in gain?*

The decrease in gain is consistent with the decrease in input resistance, especially if spike threshold does not change. More current is needed to get the neuron to spike over development, consistent with the decrease in resistance. Therefore, we did not speculate on this result in the manuscript.

*13) The following statement is unclear: “Although it is clear, from the parameter values of the centroids of each class, that most parameters do not follow a similar evolution during development (CVISI decreases monophasically while AP rise slope and AP amplitude present biphasic changes)*.*” Please clarify the meaning of this.*

There was a syntax error in this sentence, which has now been fixed:

“Based on the parameter values of the centroids of each class, it is clear that most parameters do not follow a similar trajectory during development (CVISI decreases monophasically while AP rise slope and AP amplitude present biphasic changes).”

*14) To investigate the biophysical basis underlying the 'class 1-to-class 2' transition and the 'class 2-to-class 3' transition, very different apamin concentrations were used (100 nM vs*. *5 nM, respectively). Is this motivated by (or does this correspond to) quantitative differences in AHP amplitude changes between the two transitions?*

Yes, the apamin concentrations of 100 nM and 5 nM were chosen to respectively induce a 75-80% and 20-25% block of the AHP, based on the dose-response performed by Wolfart and colleagues (J. Neurosci., 2001). These choices were motivated by the changes in AHP amplitude observed between the different developmental stages that we tried to mimic. There is a massive increase in AHP between P2 and P5, therefore the reverse transition was simulated by blocking most of the AHP with 100 nM. The more modest increase in AHP observed between P8 and P14 required to use a lower concentration (5 nM). We have added explanations in the text to make these choices clearer to the reader:

“Therefore, P7-P8 neurons were recorded and apamin application (100 nM, to block 75-80% of the AHP, Wolfart et al., 2001).”

And

“...we tested the effect of reducing AHP amplitude (using 5nM apamin to block 20- 25% of the AHP, Wolfart et al., 2001) and reducing the sodium current involved in the AP (using 20 nM TTX).”

*15) The authors provide clear evidence that immature DAergic neurons burst, and that the transition from bursting to irregular and then regular firing is a critical aspect of the electrophysiological development of the neurons. However, no reason is given why the bursts early on might be important*. *Do these bursts coincide with a period of heightened synaptogenesis and/or plasticity in these neurons?*

Although there is no clear evidence that intrinsic bursting might have a physiological relevance, we have added a short paragraph in the Discussion to speculate on its potential role:

“To date however, the physiological significance of this intrinsic bursting is unknown. In several systems including the retina and the spinal cord (Moody, Phyiol. Rev. 2005), spontaneous waves of activity early in development have been shown to participate in network development/synaptic refinement. Calcium is a central actor of this process, and the patterns of activity observed in these systems (synchronized bursts of action potentials) promote intracellular calcium rises (Moody, Physiol. Rev. 2005). While midbrain dopaminergic neurons already project to the striatum at embryonic stage E16 and release dopamine at E18 (Ferrari, Front. Cell. Neurosci. 2012; Hu, J. Comp. Neurol. 2004}, the refinement of their projections continues at least until P0 (Hu, J. Comp. Neurol. 2004), and it is likely that this pruning involves calcium-dependent mechanisms. The intrinsic bursting observed in SNc dopaminergic neurons at early postnatal stages may promote this process.”

*16) In addition to different internal EGTA concentrations, the time after going whole-cell can drastically affect firing patterns (e.g., see:*
Figure 1
*of*
*Liu and Shipley, 2008**)*. *How stable were bursting activity patterns? That is, did you observe washout of bursting?*

We agree with Reviewer 2 that whole-cell recordings can be associated with significant alterations of firing. Although we did observe quantitative changes in firing over the recording time (the spontaneous frequency of pacemaking cells typically decreases over time, although not always significantly), we did not observed qualitative changes in firing, i.e. bursting cells kept bursting throughout the recording. Most often, the bursts increased in duration and decreased in frequency, an effect likely due to the perfusion of EGTA and its blockade of calcium-activated conductances (>15-20 minutes of recording).

Reviewer #3:

*This manuscript is a great example of careful experimental design and execution, as well as presentation of the results. The figures are beautiful in their content and clarity, and the data are extremely convincing. I cannot say the same about the Title, which I find boring and unexciting. My enthusiasm is somewhat deflated by the authors' general presentation of the work, however. As it reads, I see a methods paper, while the conceptual novelty in it is underemphasized: you have applied these sophisticated analysis methods to a high-dimensional characterization of neuronal activity, and with this you have established 3 basic post-natal developmental stages. As the Discussion states, these were basically known, but you have refined the transition points and the features that best characterize these states. From these, and here is the most novel part from the biological point of view perhaps, 3 ionic currents have emerged as the best candidates to explain these transitions. I think that this is important work, but as I said, this should be portrayed in a broader light. The prediction from this analysis that multiple currents are involved in this transition is important but relegated to the end of the Abstract and as a secondary point (“This approach also predicts...”) and discussion. Previous work from the senior author suggesting the importance of recording many parameters from each neuron or preparation given the large variability observed in each parameter is relevant here but not mentioned. Isn't that the context where this work comes from? Also, I assume that you would predict that if I recorded from 300 adult basal forebrain cells, or leech interneurons (or any preparation), I should be able to extract the existence of subtypes that may otherwise not be known, or are only assumed to exist based on less rigorous protocols. It would be important to indicate this explicitly*.

We agree with Reviewer 3 that the discussion originally addressed specific issues about the SNc, instead of emphasizing the general interest of the approach presented in the study for neuroscientists from different fields. We have significantly modified the discussion to take these suggestions into account. We shortened the SNc-specific parts of the discussion, and added a section addressing the link between phenotype variability, variability of underlying conductances, and their link with phenotype definition. We have also added a new Figure (Figure 9) dedicated to the analysis of phenotypic variability across development.

*1) The first 2 sentences (ending in (*[26]*)) are rather trivial and summarized well enough in the following sentence*.

We agree with Reviewer 3 and have removed the first two sentences. We slightly modified the following two sentences, which now start the Introduction:

“The morphology and assortment of voltage-dependent and–independent conductances displayed by one particular neuronal type provide it with specific passive and active properties (26; 29). In turn, passive and active properties define the way the neuron produces or processes information, i.e. its electrical phenotype”

*2) The broad importance of this work should be expanded upon in the first paragraph of the Introduction*.

The first paragraph of the Introduction has been modified according to several suggestions of all Reviewers, and is now shorter and more focused on the central question of the study. The broad importance of the work has been mainly expanded upon in the Discussion.

*3) Description of activity types is a bit ambiguous and not consistently used: Regular tonic and pacemaker are the same thing. Irregular is irregular spiking. All of these are spontaneous. If I am right, I suggest you stick with one term throughout*.

We agree with Reviewer 3 that the multiplication of terms can be confusing, and we have tried to limit the number of terms describing activity throughout the manuscript. However, one part describes in vivo patterns of activity, i.e. with intact synaptic inputs; therefore we cannot really use spontaneous activity, as these activity patterns might be dependent on the activity of synaptic contacts. Moreover, pacemaker is a term commonly used when referring to dopaminergic neuron activity, and it is difficult to avoid it when describing the tonic regular activity pattern of activity these neurons display *in vitro*. Nevertheless, we have modified the text to limit the use of this term and favored regular tonic, which is less confusing for non- specialists.

4) What is the relationship between DA release and activation of IEGs?

The type of DA release (diffuse release *vs* “real” synaptic release) and the released concentration have been shown to be conditioned by the activity pattern of dopaminergic neurons. High concentrations of DA are only released during bursts of activity of dopaminergic neurons, and the activation of immediate early genes such as c-fos or Arc are dependent on a strong activation of second messenger pathways only triggered by high DA levels (Chergui et al, Neuroscience 1996; Howard et al., J. Neurochemistry 2013).

5) What does “strongly evolves” mean?

To avoid confusion, we replaced “strongly evolves” with “changes”.

*6) Throughout the word “evolution” is used when it seem that “development” would be more appropriate*.

We agree with Reviewer 3 and have replaced the term “evolution” with “development” or “changes” whenever it was more appropriate.

7) What is meant by “intrinsic electrophysiological maturity”?

Since no synaptic property was evaluated during this study (experiments were performed in the presence of synaptic blockers), we thought that it was more appropriate to talk about “intrinsic electrophysiological maturity” rather than “electrophysiological maturity”, which might involve maturation of synaptic inputs.

8) What is the statistical test used here?

Unpaired t-test was used in the first two instances (comparison of CVISI and AHP amplitude between 10 EGTA and 0.5 EGTA conditions) while a paired t-test was used to compare the ISIavg in cell-attached and whole-cell configuration (the values are coming from the same cells, therefore the paired test). We have added this information to the text.

*9) Speaking of conductance density is reduces clarity (and “conductances' density per surface area” is redundant). I think that it is clearer to say that capacitance and leak increase, which results in a constant tau_m*.

We agree with Reviewer 3 that conductance density per surface area is redundant and have modified the text accordingly (by removing “per surface area”). This sentence now reads as follows:

“These data suggest that, after P2-3, membrane surface area increases but that leak conductances’ density stays fairly constant, such that the measured input conductance (1/R_in_) scales with Cm (since Cm scales with membrane surface).”

*10) Why did you choose to only use those 8 parameters and not more? Also, as you data in*
Figure 5
*show, I would argue that the AP amplitude and AP rise slope are tightly linked (dV/dt and AP amp are both dependent on I_Na). Likewise with AP half-width and AP decay slope (*Figure 5
*are nearly identical). Would your conclusions still hold if two parameters (one from each of these pairs of related parameters) were removed from theanalyzes? I should think not since they seem to be redundant*.

We agree with Reviewer 3 regarding the possible redundancy between some of the electrophysiological properties that were analyzed. AP rise slope and AP amplitude are definitely correlated (although not with an r^2^ of 1), and AP half- width and decay slope are also correlated (although again not perfectly). Our original purpose was to include as many discrete properties as possible to ensure that no meaningful parameter was omitted, therefore the redundancy (but again not full redundancy) between some of the properties included in the analysis.

As detailed above in major point 2 from Reviewer 1: concerning the 8 parameters chosen for the PCA, the choice was made based on the recording condition needed to measure these 8 parameters: we specifically selected the 8 parameters that can be extracted from recordings of spontaneous activity, which do not involve any current injection. The basic question was: is it possible to accurately define electrical phenotype based on non-manipulated recordings of spontaneous activity? We have modified the sentence introducing the multi- variate analyses to better explain the rationale for using only these 8 specific parameters in the clustering and PCA:

“We chose to include only the 8 electrophysiological parameters that can be extracted from the recordings of spontaneous activity (ISI_avg_, CV_ISI_, AP threshold, AP amplitude, AP rise slope, AP decay slope, AP half-width and AHP amplitude), and excluded the parameters that require current injection to be measured (R_in_, C_m_, τ_m_, sag amplitude, rebound delay, GS, GE, SFA index), the goal being to determine whether phenotype can be accurately defined with minimal electrophysiological manipulation (no current injection).”

*11) CV_ISI are said to be different and*
Figure 7
*is referred to. I see a difference but this is not with quantitative certainty*.

The differences in CVISI are very significant either when we compare the central objects of each class (colored traces in 7B) or when we perform a comparison between the average values of each class. The three neurons presented in Figure 7 as the central objects of class 1, 2 and 3 have CVISI values of 133%, 48% and 12%, respectively. The average values for the same classes are the ones shown in the table 7C (143.53, 45.91 and 10.69, respectively), and are also statistically different (one-way ANOVA on ranks, p<0.05).

*12) To speak of “anti-conductance” sounds very strange. It is simply a negative conductance and thus used to subtract leak*.

We agree with Reviewer 3 and have replaced “anti-conductance” by “negative conductance” in the text, and also “anti-leak” by “leak subtraction” where appropriate.

*13) 5nM apamin is used while for P7-P8 cells then100nM was used*. *Is this correct?*

Yes, the apamin concentrations of 100 nM and 5 nM were chosen to respectively induce a 75-80% and 20-25% block of the AHP, based on the dose-response performed by Wolfart and colleagues (J. Neurosci., 2001). These choices were motivated by the changes in AHP amplitude observed between the different developmental stages that we tried to mimic. There is a massive increase in AHP between P2 and P5, therefore the reverse transition was simulated by blocking most of the AHP with 100 nM. The more modest increase in AHP observed between P8 and P14 necessitated to use a lower concentration. We have added explanations in the text to make these choices clearer to the reader:

“Therefore, P7-P8 neurons were recorded and apamin application (100 nM, to block 75-80% of the AHP, Wolfart et al., 2001).”

And

“...we tested the effect of reducing AHP amplitude (using 5nM apamin to block 20- 25% of the AHP, Wolfart et al., 2001) and reducing the sodium current involved in the AP (using 20 nM TTX).”

*14) Table 2 and elsewhere*. *Isn't your statistical stacking method similar or equivalent to Taylor's clutter-based dimension reordering? (*[59]*)*

We thank Reviewer 3 for pointing out this regrettable oversight. The clutter- based dimension reordering and dimensional stacking used in Adam Taylor’s work was definitely an inspiration for the statistical stacking presented in our study, and should have been properly cited as such. However, our method is much more limited and less sophisticated than the method developed by [59]. We have added a sentence to give the proper credit to Taylor’s study in the Discussion:

“... we propose an intermediate solution which consists of a visual stacking of the results of the uni-variate analyses performed on each parameter. Dimensional stacking consists in visualizing a high-dimensional data set in a two-dimension space, and has been recently applied to the analysis of the electrical behavior of databases of realistic neuron models (Taylor, J. Neurophysiol. 2006). ”

*15) Table 2. I believe that using Rin, Cm and Tau_m in the same statistical comparison is not valid since Tau_m depends on the other two parameters*.

We agree that there is redundancy in using Rin, Cm and Tau_m. However Table 2 has several roles, and is not only intended to provide a general overview of phenotype development (using the stacking of the statistical results). In fact, its first role was to summarize the results of the univariate statistical analyses, such that the reader could follow the changes in any of the 16 parameters analyzed. As such, excluding Rin, Cm or Tau_m would have compromised the goal of using table 2 as a statistical summary table.

“... we propose an intermediate solution which consists of a visual stacking of the results of the uni-variate analyses performed on each parameter. Dimensional stacking consists in visualizing a high-dimensional data set in a two-dimension space, and has been recently applied to the analysis of the electrical behavior of databases of realistic neuron models (Taylor, J. Neurophysiol. 2006).”